# Calibrating Deep Neural Networks using Focal Loss

**Jishnu Mukhoti**[*]
University of Oxford
FiveAI Ltd.

**Viveka Kulharia**[*]
University of Oxford

**Amartya Sanyal**
University of Oxford
The Alan Turing Institute

**Stuart Golodetz**
FiveAI Ltd.

**Philip H. S. Torr**
University of Oxford
FiveAI Ltd.

**Puneet K. Dokania**
University of Oxford
FiveAI Ltd.

## Abstract

Miscalibration – a mismatch between a model's confidence and its correctness – of Deep Neural Networks (DNNs) makes their predictions hard to rely on. Ideally, we want networks to be accurate, calibrated and confident. We show that, as opposed to the standard cross-entropy loss, focal loss [19] allows us to learn models that are already very well calibrated. When combined with temperature scaling, whilst preserving accuracy, it yields state-of-the-art calibrated models. We provide a thorough analysis of the factors causing miscalibration, and use the insights we glean from this to justify the empirically excellent performance of focal loss. To facilitate the use of focal loss in practice, we also provide a principled approach to automatically select the hyperparameter involved in the loss function. We perform extensive experiments on a variety of computer vision and NLP datasets, and with a wide variety of network architectures, and show that our approach achieves state-of-the-art calibration without compromising on accuracy in almost all cases. Code is available at `https://github.com/torrvision/focal_calibration`.

## 1 Introduction

Deep neural networks have dominated computer vision and machine learning in recent years, and this has led to their widespread deployment in real-world systems [2, 3, 11, 12, 38]. However, many current multi-class classification networks in particular are poorly calibrated, in the sense that the probability values that they associate with the class labels they predict overestimate the likelihoods of those class labels being correct in the real world. This is a major problem, since if networks are routinely overconfident, then downstream components cannot trust their predictions. The underlying cause is hypothesised to be that these networks' high capacity leaves them vulnerable to overfitting on the negative log-likelihood (NLL) loss they conventionally use during training [7].

Given the importance of this problem, numerous suggestions for how to address it have been proposed. Much work has been inspired by approaches that were not originally formulated in a deep learning context, such as Platt scaling [30], histogram binning [40], isotonic regression [41], and Bayesian binning and averaging [22, 21]. As deep learning has become more dominant, however, various works have begun to directly target the calibration of deep networks. For example, Guo et al. [7] have popularised a modern variant of Platt scaling known as *temperature scaling*, which works by dividing a network's logits by a scalar $T > 0$ (learnt on a validation subset) prior to performing softmax. Temperature scaling has the desirable property that it can improve the calibration of a network without in any way affecting its accuracy. However, whilst its simplicity and effectiveness

---

[*]Joint first authors, order decided by coin flip. Contact: {jishnu, viveka, puneet, phst}@robots.ox.ac.uk, amartya.sanyal@cs.ox.ac.uk, sgolodetz@gxstudios.net

have made it a popular network calibration method, it does have downsides. For example, whilst it scales the logits to reduce the network's confidence in incorrect predictions, this also slightly reduces the network's confidence in predictions that were correct [16]. Moreover, it is known that temperature scaling does not calibrate a model under data distribution shift [27].

By contrast, [16] initially eschew temperature scaling in favour of minimising a differentiable proxy for calibration error at training time, called Maximum Mean Calibration Error (MMCE), although they do later also use temperature scaling as a post-processing step to obtain better results than cross-entropy followed by temperature scaling [7]. Separately, [20] propose training models on cross-entropy loss with label smoothing instead of one-hot labels, and show that label smoothing has a very favourable effect on model calibration.

In this paper, we propose a technique for improving network calibration that works by replacing the cross-entropy loss conventionally used when training classification networks with the focal loss proposed by [19]. We observe that unlike cross-entropy, which minimises the KL divergence between the predicted (softmax) distribution and the target distribution (one-hot encoding in classification tasks) over classes, focal loss minimises a regularised KL divergence between these two distributions, which ensures minimisation of the KL divergence whilst *increasing the entropy* of the predicted distribution, thereby preventing the model from becoming overconfident. Since focal loss, as shown in §4, is dependent on a hyperparameter, $\gamma$, that needs to be cross-validated, we also provide a method for choosing $\gamma$ automatically for each sample, and show that it outperforms all the baseline models.

The intuition behind using focal loss is to direct the network's attention during training towards samples for which it is currently predicting a low probability for the correct class, since trying to reduce the NLL on samples for which it is already predicting a high probability for the correct class is liable to lead to NLL overfitting, and thereby miscalibration [7]. More formally, we show in §4 that focal loss can be seen as *implicitly* regularising the weights of the network during training by causing the gradient norms for confident samples to be lower than they would have been with cross-entropy, which we would expect to reduce overfitting and improve the network's calibration.

Overall, we make the following contributions:

1. In §3, we study the link that [7] observed between miscalibration and NLL overfitting in detail, and show that the overfitting is associated with the predicted distributions for misclassified test samples becoming peakier as the optimiser tries to increase the magnitude of the network's weights to reduce the training NLL.
2. In §4, we propose the use of focal loss for training better-calibrated networks, and provide both theoretical and empirical justifications for this approach. In addition, we provide a principled method for automatically choosing $\gamma$ for each sample during training.
3. In §5, we show, via experiments on a variety of classification datasets and network architectures, that DNNs trained with focal loss are more calibrated than those trained with cross-entropy loss (both with and without label smoothing), MMCE or Brier loss [1]. Finally, we also make the interesting observation that whilst temperature scaling may not work for detecting out-of-distribution (OoD) samples, our approach can. We show that our approach is better at detecting out-of-distribution samples, taking CIFAR-10 as the in-distribution dataset, and SVHN and CIFAR-10-C as out-of-distribution datasets.

## 2  Problem Formulation

Let $D = \langle (\mathbf{x}_i, y_i) \rangle_{i=1}^{N}$ denote a dataset consisting of $N$ samples from a joint distribution $\mathcal{D}(\mathcal{X}, \mathcal{Y})$, where for each sample $i$, $\mathbf{x}_i \in \mathcal{X}$ is the input and $y_i \in \mathcal{Y} = \{1, 2, ..., K\}$ is the ground-truth class label. Let $\hat{p}_{i,y} = f_\theta(y|\mathbf{x}_i)$ be the probability that a neural network $f$ with model parameters $\theta$ predicts for a class $y$ on a given input $\mathbf{x}_i$. The class that $f$ predicts for $\mathbf{x}_i$ is computed as $\hat{y}_i = \mathrm{argmax}_{y \in \mathcal{Y}} \, \hat{p}_{i,y}$, and the predicted confidence as $\hat{p}_i = \max_{y \in \mathcal{Y}} \, \hat{p}_{i,y}$. The network is said to be *perfectly calibrated* when, for each sample $(\mathbf{x}, y) \in D$, the confidence $\hat{p}$ is equal to the model accuracy $\mathbb{P}(\hat{y} = y|\hat{p})$, i.e. the probability that the predicted class is correct. For instance, of all the samples to which a perfectly calibrated neural network assigns a confidence of $0.8$, $80\%$ should be correctly predicted.

A popular metric used to measure model calibration is the *expected calibration error* (ECE) [22], defined as the expected absolute difference between the model's confidence and its accuracy, i.e. $\mathbb{E}_{\hat{p}}\big[\,|\mathbb{P}(\hat{y} = y|\hat{p}) - \hat{p}|\,\big]$. Since we only have finite samples, the ECE cannot in practice be computed using this definition. Instead, we divide the interval $[0, 1]$ into $M$ equispaced bins, where the $i^{\text{th}}$ bin

is the interval $\left(\frac{i-1}{M}, \frac{i}{M}\right]$. Let $B_i$ denote the set of samples with confidences belonging to the $i^{\text{th}}$ bin. The accuracy $A_i$ of this bin is computed as $A_i = \frac{1}{|B_i|} \sum_{j \in B_i} \mathbb{1}(\hat{y}_j = y_j)$, where $\mathbb{1}$ is the indicator function, and $\hat{y}_j$ and $y_j$ are the predicted and ground-truth labels for the $j^{\text{th}}$ sample. Similarly, the confidence $C_i$ of the $i^{\text{th}}$ bin is computed as $C_i = \frac{1}{|B_i|} \sum_{j \in B_i} \hat{p}_j$, i.e. $C_i$ is the average confidence of all samples in the bin. The ECE can be approximated as a weighted average of the absolute difference between the accuracy and confidence of each bin: $\text{ECE} = \sum_{i=1}^{M} \frac{|B_i|}{N} |A_i - C_i|$.

A similar metric, the *maximum calibration error* (MCE) [22], is defined as the maximum absolute difference between the accuracy and confidence of each bin: $\text{MCE} = \max_{i \in \{1,...,M\}} |A_i - C_i|$.

**AdaECE:** One disadvantage of ECE is the uniform bin width. For a trained model, most of the samples lie within the highest confidence bins, and hence these bins dominate the value of the ECE. We thus also consider another metric, AdaECE (Adaptive ECE), for which bin sizes are calculated so as to evenly distribute samples between bins (similar to the adaptive binning procedure in [25]): $\text{AdaECE} = \sum_{i=1}^{M} \frac{|B_i|}{N} |A_i - C_i|$ s.t. $\forall i, j \cdot |B_i| = |B_j|$.

**Classwise-ECE:** The ECE metric only considers the probability of the predicted class, without considering the other scores in the softmax distribution. A stronger definition of calibration would require the probabilities of all the classes in the softmax distribution to be calibrated [14, 36, 39, 15]. This can be achieved with a simple classwise extension of the ECE metric: $\text{ClasswiseECE} = \frac{1}{K} \sum_{i=1}^{M} \sum_{j=1}^{K} \frac{|B_{i,j}|}{N} |A_{i,j} - C_{i,j}|$, where $K$ is the number of classes, $B_{ij}$ denotes the set of samples from the $j^{th}$ class in the $i^{th}$ bin, $A_{ij} = \frac{1}{|B_{ij}|} \sum_{k \in B_{ij}} \mathbb{1}(j = y_k)$ and $C_{i,j} = \frac{1}{|B_{ij}|} \sum_{k \in B_{ij}} \hat{p}_{kj}$.

A common way of visualising calibration is to use a *reliability plot* [26], which plots the accuracies of the confidence bins as a bar chart (see Appendix Figure A.1). For a perfectly calibrated model, the accuracy for each bin matches the confidence, and hence all of the bars lie on the diagonal. By contrast, if most of the bars lie above the diagonal, the model is more accurate than it expects, and is under-confident, and if most of the bars lie below the diagonal, then it is over-confident.

# 3   What Causes Miscalibration?

We now discuss why high-capacity neural networks, despite achieving low classification errors on well-known datasets, tend to be miscalibrated. A key empirical observation made by [7] was that poor calibration of such networks appears to be linked to overfitting on the negative log-likelihood (NLL) during training. In this section, we further inspect this observation to provide new insights.

For the analysis, we train a ResNet-50 network on CIFAR-10 with state-of-the-art performance settings [31]. We use Stochastic Gradient Descent (SGD) with a mini-batch of size 128, momentum of 0.9, and learning rate schedule of $\{0.1, 0.01, 0.001\}$ for the first 150, next 100, and last 100 epochs, respectively. We minimise cross-entropy loss (a.k.a. NLL) $\mathcal{L}_c$, which, in a standard classification context, is $-\log \hat{p}_{i,y_i}$, where $\hat{p}_{i,y_i}$ is the probability assigned by the network to the correct class $y_i$ for the $i^{th}$ sample. Note that the NLL is minimised when for each training sample $i$, $\hat{p}_{i,y_i} = 1$, whereas the classification error is minimised when $\hat{p}_{i,y_i} > \hat{p}_{i,y}$ for all $y \neq y_i$. This indicates that even when the classification error is 0, the NLL can be positive, and the optimisation algorithm can still try to reduce it to 0 by further increasing the value of $\hat{p}_{i,y_i}$ for each sample (see Appendix A).

To study how miscalibration occurs during training, we plot the average NLL for the train and test sets at each training epoch in Figures 1(a) and 1(b). We also plot the average NLL and the entropy of the softmax distribution produced by the network for the correctly and incorrectly classified samples. In Figure 1(c), we plot the classification errors on the train and test sets, along with the test set ECE.

**Curse of misclassified samples:** Figures 1(a) and 1(b) show that although the average train NLL (for both correctly and incorrectly classified training samples) broadly decreases throughout training, after the $150^{th}$ epoch (where the learning rate drops by a factor of 10), there is a marked rise in the average test NLL, indicating that the network starts to overfit on average NLL. This increase in average test NLL is caused only by the incorrectly classified samples, as the average NLL for the correctly classified samples continues to decrease even after the $150^{th}$ epoch. We also observe that after epoch 150, the test set ECE rises, indicating that the network is becoming miscalibrated. This corroborates the observation in [7] that miscalibration and NLL overfitting are linked.

**Peak at the wrong place:** We further observe that the entropies of the softmax distributions for both the correctly and incorrectly classified *test* samples decrease throughout training (in other words, the

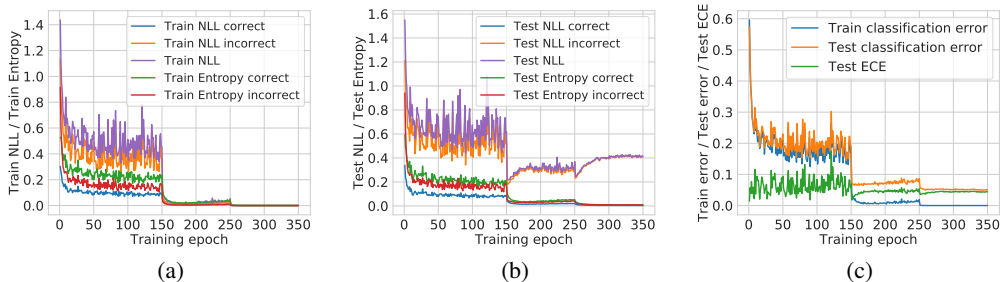

Figure 1: Metrics related to calibration plotted whilst training a ResNet-50 network on CIFAR-10.

distributions get peakier). This observation, coupled with the one we made above, indicates that *for the wrongly classified test samples, the network gradually becomes more and more confident about its incorrect predictions*.

**Weight magnification:** The increase in confidence of the network's predictions can happen if the network increases the norm of its weights $W$ to increase the magnitudes of the logits. In fact, cross-entropy loss is minimised when for each training sample $i$, $\hat{p}_{i,y_i} = 1$, which is possible only when $||W|| \to \infty$. Cross-entropy loss thus inherently induces this tendency of weight magnification in neural network optimisation. The promising performance of weight decay [7] (regulating the norm of weights) on the calibration of neural networks can perhaps be explained using this. This increase in the network's confidence during training is one of the key causes of miscalibration.

## 4   Improving Calibration using Focal Loss

As discussed in §3, overfitting on NLL, which is observed as the network grows more confident on all of its predictions irrespective of their correctness, is strongly related to poor calibration. One cause of this is that the cross-entropy objective minimises the difference between the softmax distribution and the ground-truth one-hot encoding over an entire mini-batch, irrespective of how well a network classifies individual samples in the mini-batch. In this work, we study an alternative loss function, popularly known as *focal loss* [19], that tackles this by weighting loss components generated from individual samples in a mini-batch by how well the model classifies them. For classification tasks where the target distribution is a one-hot encoding, it is defined as $\mathcal{L}_f = -(1 - \hat{p}_{i,y_i})^\gamma \log \hat{p}_{i,y_i}$, where $\gamma$ is a user-defined hyperparameter[2].

**Why might focal loss improve calibration?** We know that cross-entropy forms an upper bound on the KL-divergence between the target distribution $q$ and the predicted distribution $\hat{p}$, i.e. $\mathcal{L}_c \geq \mathrm{KL}(q||\hat{p})$, so minimising cross-entropy results in minimising $\mathrm{KL}(q||\hat{p})$. Interestingly, a general form of focal loss can be shown to be an upper bound on the regularised KL-divergence, where the regulariser is the negative entropy of the predicted distribution $\hat{p}$, and the regularisation parameter is $\gamma$, the hyperparameter of focal loss (a proof of this can be found in Appendix B):

$$\mathcal{L}_f \geq \mathrm{KL}(q||\hat{p}) - \gamma \mathbb{H}[\hat{p}]. \tag{1}$$

The most interesting property of this upper bound is that it shows that replacing cross-entropy with focal loss has the effect of adding a maximum-entropy regulariser [29] to the implicit minimisation that was previously being performed. In other words, trying to minimise focal loss minimises the KL divergence between $\hat{p}$ and $q$, whilst simultaneously increasing the entropy of the predicted distribution $\hat{p}$. Note, in the case of ground truth with one-hot encoding, only the component of the entropy of $\hat{p}$ corresponding to the ground-truth index, $\gamma(-\hat{p}_{i,y_i} \log \hat{p}_{i,y_i})$, will be maximised (refer Appendix B). Encouraging the predicted distribution to have higher entropy can help avoid the overconfident predictions produced by DNNs (see the 'Peak at the wrong place' paragraph of §3), and thereby improve calibration.

**Empirical observations:** To analyse the behaviour of neural networks trained on focal loss, we use the same framework as mentioned above, and train four ResNet-50 networks on CIFAR-10, one using cross-entropy loss, and three using focal loss with $\gamma = 1, 2$ and $3$. Figure 2(a) shows that the test NLL

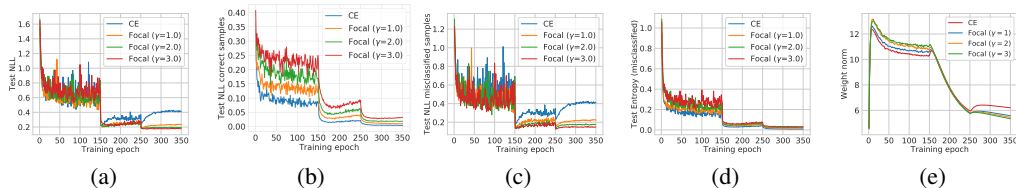

Figure 2: How metrics related to model calibration change whilst training several ResNet-50 networks on CIFAR-10, using either cross-entropy loss, or focal loss with $\gamma$ set to 1, 2 or 3.

for the cross-entropy model significantly increases towards the end of training (before saturating), whereas the NLLs for the focal loss models remain low. To better understand this, we analyse the behaviour of these models for correctly and incorrectly classified samples. Figure 2(b) shows that even though the NLLs for the correctly classified samples broadly-speaking decrease over the course of training for all the models, the NLLs for the focal loss models remain consistently higher than that for the cross-entropy model throughout training, implying that the focal loss models are relatively less confident than the cross-entropy model for samples that they predict correctly. This is important, as we have already discussed that it is overconfidence that normally makes deep neural networks miscalibrated. Figure 2(c) shows that in contrast to the cross-entropy model, for which the NLL for misclassified test samples increases significantly after epoch 150, the rise in this value for the focal loss models is much less severe. Additionally, in Figure 2(d), we notice that the entropy of the softmax distribution for misclassified test samples is consistently (if marginally) higher for focal loss than for cross-entropy (consistent with Equation 1).

Note that from Figure 2(a), one may think that applying early stopping when training a model on cross-entropy can provide better calibration scores. However, there is no ideal way of doing early stopping that provides the best calibration error and the best test set accuracy. For fair comparison, we chose 3 intermediate models for each loss function with the best val set ECE, NLL and accuracy, and observed that: a) for every stopping criterion, focal loss outperforms cross-entropy in both test set accuracy and ECE, b) when using val set ECE as a stopping criterion, the intermediate model for cross-entropy indeed improves its test set ECE, but at the cost of a significantly higher test error. Please refer to Appendix J for more details.

As per §3, an increase in the test NLL and a decrease in the test entropy for misclassified samples, along with no corresponding increase in the test NLL for the correctly classified samples, can be interpreted as the network starting to predict softmax distributions for the misclassified samples that are ever more peaky in the wrong place. Notably, our results in Figures 2(b), 2(c) and 2(d) clearly show that this effect is significantly reduced when training with focal loss rather than cross-entropy, leading to a better-calibrated network whose predictions are less peaky in the wrong place.

**Theoretical justification:** As mentioned previously, once a model trained using cross-entropy reaches high training accuracy, the optimiser may try to further reduce the training NLL by increasing the confidences for the correctly classified samples. It may achieve this by magnifying the network weights to increase the magnitudes of the logits. To verify this hypothesis, we plot the $L_2$ norm of the weights of the last linear layer for all four networks as a function of the training epoch (see Figure 2(e)). Notably, although the norms of the weights for the models trained on focal loss are initially higher than that for the cross-entropy model, *a complete reversal* in the ordering of the weight norms occurs between epochs 150 and 250. In other words, as the networks start to become miscalibrated, the weight norm for the cross-entropy model also starts to become greater than those for the focal loss models. In practice, this is because focal loss, by design, starts to act as a regulariser on the network's weights once the model has gained a certain amount of confidence in its predictions. This behaviour of focal loss can be observed even on a much simpler setup like a linear model (see Appendix C). To better understand this, we start by considering the following proposition (proof in Appendix D):

**Proposition 1.** *For focal loss $\mathcal{L}_f$ and cross-entropy $\mathcal{L}_c$, the gradients $\frac{\partial \mathcal{L}_f}{\partial \mathbf{w}} = \frac{\partial \mathcal{L}_c}{\partial \mathbf{w}} g(\hat{p}_{i,y_i}, \gamma)$, where $g(p, \gamma) = (1-p)^\gamma - \gamma p (1-p)^{\gamma-1} \log(p)$, $\gamma \in \mathbb{R}^+$ is the focal loss hyperparameter, and $\mathbf{w}$ denotes the parameters of the last linear layer. Thus $\left\| \frac{\partial \mathcal{L}_f}{\partial \mathbf{w}} \right\| \leq \left\| \frac{\partial \mathcal{L}_c}{\partial \mathbf{w}} \right\|$ if $g(\hat{p}_{i,y_i}, \gamma) \in [0, 1]$.*

Proposition 1 shows the relationship between the norms of the gradients of the last linear layer for focal loss and cross-entropy loss, for the same network architecture. Note that this relation depends on a function $g(p, \gamma)$, which we plot in Figure 3(a) to understand its behaviour. It is clear that for

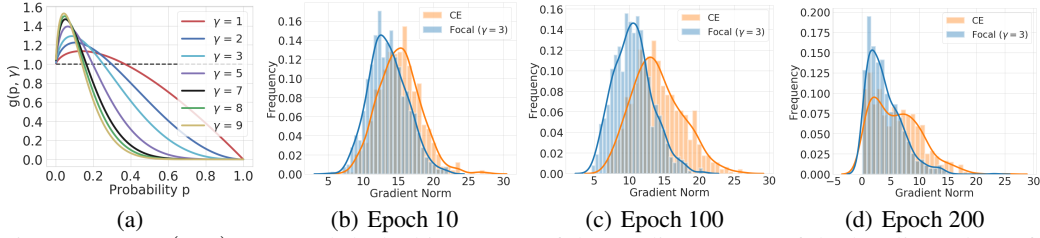

(a)  (b) Epoch 10  (c) Epoch 100  (d) Epoch 200

Figure 3: (a): $g(p,\gamma)$ vs. $p$ and (b-d): histograms of the gradient norms of the last linear layer for both cross-entropy and focal loss.

every $\gamma$, there exists a (different) threshold $p_0$ such that for all $p \in [0, p_0]$, $g(p,\gamma) \geq 1$, and for all $p \in (p_0, 1]$, $g(p,\gamma) < 1$. (For example, for $\gamma = 1$, $p_0 \approx 0.4$.) We use this insight to further explain why focal loss provides implicit weight regularisation.

**Implicit weight regularisation:** For a network trained using focal loss with a fixed $\gamma$, during the initial stages of the training, when $\hat{p}_{i,y_i} \in (0, p_0)$, $g(\hat{p}_{i,y_i}, \gamma) > 1$. This implies that the confidences of the focal loss model's predictions will initially increase faster than they would for cross-entropy. However, as soon as $\hat{p}_{i,y_i}$ crosses the threshold $p_0$, $g(\hat{p}_{i,y_i}, \gamma)$ falls below 1 and reduces the size of the gradient updates made to the network weights, thereby having a regularising effect on the weights. This is why, in Figure 2(e), we find that the weight norms of the models trained with focal loss are initially higher than that for the model trained using cross-entropy. However, as training progresses, we find that the ordering of the weight norms reverses, as focal loss starts regularising the network weights. Moreover, we can draw similar insights from Figures 3(b), 3(c) and 3(d), in which we plot histograms of the gradient norms of the last linear layer (over all samples in the training set) at epochs 10, 100 and 200, respectively. At epoch 10, the gradient norms for cross-entropy and focal loss are similar, but as training progresses, those for cross-entropy decrease less rapidly than those for focal loss, indicating that the gradient norms for focal loss are consistently lower than those for cross-entropy throughout training.

Finally, observe in Figure 3(a) that for higher $\gamma$ values, the fall in $g(p,\gamma)$ is steeper. We would thus expect a greater weight regularisation effect for models that use higher values of $\gamma$. This explains why, of the three models that we trained using focal loss, the one with $\gamma = 3$ outperforms (in terms of calibration) the one with $\gamma = 2$, which in turn outperforms the model with $\gamma = 1$. Based on this observation, one might think that, in general, a higher value of gamma would lead to a more calibrated model. However, this is not the case, as we notice from Figure 3(a) that for $\gamma \geq 7$, $g(p,\gamma)$ reduces to nearly 0 for a relatively low value of $p$ (around 0.5). As a result, using values of $\gamma$ that are too high will cause the gradients to die (i.e. reduce to nearly 0) early, at a point at which the network's predictions remain ambiguous, thereby causing the training process to fail.

**How to choose** $\gamma$: As discussed, focal loss provides implicit entropy and weight regularisation, which heavily depend on the value of $\gamma$. Finding an appropriate $\gamma$ is normally done using cross-validation. Also, traditionally, $\gamma$ is fixed for all samples in the dataset. However, as shown, the regularisation effect for a sample $i$ depends on $\hat{p}_{i,y_i}$, i.e. the predicted probability for the ground truth label for the sample. It thus makes sense to choose $\gamma$ for each sample based on the value of $\hat{p}_{i,y_i}$. To this end, we provide Proposition 2 (proof in Appendix D), which we use to find a solution to this problem:

**Proposition 2.** *Given a $p_0$, for $1 \geq p \geq p_0 > 0$, $g(p,\gamma) \leq 1$ for all $\gamma \geq \gamma^* = \frac{a}{b} + \frac{1}{\log a} W_{-1} \big( - \frac{a^{(1-a/b)}}{b} \log a \big)$, where $a = 1 - p_0$, $b = p_0 \log p_0$, and $W_{-1}$ is the Lambert-W function [4]. Moreover, for $p \geq p_0 > 0$ and $\gamma \geq \gamma^*$, the equality $g(p,\gamma) = 1$ holds only for $p = p_0$ and $\gamma = \gamma^*$.*

It is worth noting that there exist multiple values of $\gamma$ where $g(p,\gamma) \leq 1$ for all $p \geq p_0$. For a given $p_0$, Proposition 2 allows us to compute $\gamma$ s.t. (i) $g(p_0, \gamma) = 1$; (ii) $g(p,\gamma) \geq 1$ for $p \in [0, p_0)$; and (iii) $g(p,\gamma) < 1$ for $p \in (p_0, 1]$. This allows us to control the magnitude of the gradients for a particular sample $i$ based on the current value of $\hat{p}_{i,y_i}$, and gives us a way of obtaining an informed value of $\gamma$ for each sample. For instance, a reasonable policy might be to choose $\gamma$ s.t. $g(\hat{p}_{i,y_i}, \gamma) > 1$ if $\hat{p}_{i,y_i}$ is small (say less than 0.25), and $g(\hat{p}_{i,y_i}, \gamma) < 1$ otherwise. Such a policy will have the effect of making the weight updates larger for samples having a low predicted probability for the correct class and smaller for samples with a relatively higher predicted probability for the correct class.

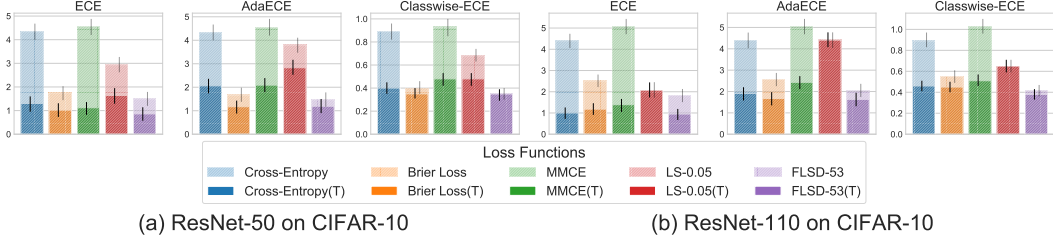

(a) ResNet-50 on CIFAR-10         (b) ResNet-110 on CIFAR-10

Figure 4: Bar plots with confidence intervals for ECE, AdaECE and Classwise-ECE, computed for ResNet-50 (first 3 figures) and ResNet-110 (last 3 figures) on CIFAR-10.

| Dataset | Model | Cross-Entropy | | Brier Loss | | MMCE | | LS-0.05 | | FL-3 (Ours) | | FLSD-53 (Ours) | |
|---|---|---|---|---|---|---|---|---|---|---|---|---|---|
| | | Pre T | Post T | Pre T | Post T | Pre T | Post T | Pre T | Post T | Pre T | Post T | Pre T | Post T |
| CIFAR-100 | ResNet-50 | 17.52 | 3.42(2.1) | 6.52 | 3.64(1.1) | 15.32 | 2.38(1.8) | 7.81 | 4.01(1.1) | 5.13 | **1.97(1.1)** | 4.5 | 2.0(1.1) |
| | ResNet-110 | 19.05 | 4.43(2.3) | **7.88** | 4.65(1.2) | 19.14 | **3.86(2.3)** | 11.02 | 5.89(1.1) | 8.64 | 3.95(1.2) | 8.56 | 4.12(1.2) |
| | Wide-ResNet-26-10 | 15.33 | 2.88(2.2) | 4.31 | 2.7(1.1) | 13.17 | 4.37(1.9) | 4.84 | 4.84(1) | **2.13** | 2.13(1) | 3.03 | **1.64(1.1)** |
| | DenseNet-121 | 20.98 | 4.27(2.3) | 5.17 | 2.29(1.1) | 19.13 | 3.06(2.1) | 12.89 | 7.52(1.2) | 4.15 | **1.25(1.1)** | 3.73 | 1.31(1.1) |
| CIFAR-10 | ResNet-50 | 4.35 | 1.35(2.5) | 1.82 | 1.08(1.1) | 4.56 | 1.19(2.6) | 2.96 | 1.67(0.9) | **1.48** | 1.42(1.1) | 1.55 | **0.95(1.1)** |
| | ResNet-110 | 4.41 | 1.09(2.8) | 2.56 | 1.25(1.2) | 5.08 | 1.42(2.8) | 2.09 | 2.09(1) | **1.55** | **1.02(1.1)** | 1.87 | 1.07(1.1) |
| | Wide-ResNet-26-10 | 3.23 | 0.92(2.2) | **1.25** | 1.25(1) | 3.29 | 0.86(2.2) | 4.26 | 1.84(0.8) | 1.69 | 0.97(0.9) | 1.56 | **0.84(0.9)** |
| | DenseNet-121 | 4.52 | 1.31(2.4) | 1.53 | 1.53(1) | 5.1 | 1.61(2.5) | 1.88 | 1.82(0.9) | 1.32 | 1.26(0.9) | **1.22** | **1.22(1)** |
| Tiny-ImageNet | ResNet-50 | 15.32 | 5.48(1.4) | 4.44 | 4.13(0.9) | 13.01 | 5.55(1.3) | 15.23 | 6.51(0.7) | 1.87 | 1.87(1) | **1.76** | **1.76(1)** |
| 20 Newsgroups | Global Pooling CNN | 17.92 | 2.39(3.4) | 13.58 | 3.22(2.3) | 15.48 | 6.78(2.2) | **4.79** | 2.54(1.1) | 8.67 | 3.51(1.5) | 6.92 | **2.19(1.5)** |
| SST Binary | Tree-LSTM | 7.37 | 2.62(1.8) | 9.01 | 2.79(2.5) | 5.03 | 4.02(1.5) | **4.84** | 4.11(1.2) | 16.05 | 1.78(0.5) | 9.19 | **1.83(0.7)** |

Table 1: ECE (%) computed for different approaches both pre and post temperature scaling (cross-validating T on ECE). Optimal temperature for each method is indicated in brackets. $T \approx 1$ indicates innately calibrated model.

Following the aforementioned arguments, we choose a threshold $p_0$ of 0.25, and use Proposition 2 to obtain a $\gamma$ policy such that $g(p, \gamma)$ is observably greater than 1 for $p \in [0, 0.25)$ and $g(p, \gamma) < 1$ for $p \in (0.25, 1]$. In particular, we use the following schedule: if $\hat{p}_{i,y_i} \in [0, 0.2)$, then $\gamma = 5$, otherwise $\gamma = 3$ (note that $g(0.2, 5) \approx 1$ and $g(0.25, 3) \approx 1$: see Figure 3(a)). We find this $\gamma$ policy to perform consistently well across multiple classification datasets and network architectures. Having said that, one can calculate multiple such schedules for $\gamma$ following Proposition 2, using the intuition of having a relatively high $\gamma$ for low values of $\hat{p}_{i,y_i}$ and a relatively low $\gamma$ for high values of $\hat{p}_{i,y_i}$.

## 5 Experiments

We conduct image and document classification experiments to test the performance of focal loss. For the former, we use CIFAR-10/100 [13] and Tiny-ImageNet [6] , and train ResNet-50, ResNet-110 [8], Wide-ResNet-26-10 [42] and DenseNet-121 [10] models, and for the latter, we use 20 Newsgroups [17] and Stanford Sentiment Treebank (SST) [32] datasets and train Global Pooling CNN [18] and Tree-LSTM [33] models. Further details on the datasets and training can be found in Appendix E.

**Baselines** Along with cross-entropy loss, we compare our method against the following baselines: a) *MMCE* (Maximum Mean Calibration Error) [16], a continuous and differentiable proxy for calibration error that is normally used as a regulariser alongside cross-entropy, b) *Brier loss* [1], the squared error between the predicted softmax vector and the one-hot ground truth encoding (Brier loss is an important baseline as it can be decomposed into calibration and refinement [5]), and c) *Label smoothing* [20] (LS): given a one-hot ground-truth distribution $\mathbf{q}$ and a smoothing factor $\alpha$ (hyperparameter), the smoothed vector $\mathbf{s}$ is obtained as $\mathbf{s}_i = (1 - \alpha)\mathbf{q}_i + \alpha(1 - \mathbf{q}_i)/(K - 1)$, where $\mathbf{s}_i$ and $\mathbf{q}_i$ denote the $i^{th}$ elements of $\mathbf{s}$ and $\mathbf{q}$ respectively, and $K$ is the number of classes. Instead of $\mathbf{q}$, $\mathbf{s}$ is treated as the ground truth. We train models using $\alpha = 0.05$ and $\alpha = 0.1$, but find $\alpha = 0.05$ to perform better. We thus report the results obtained from LS-0.05 with $\alpha = 0.05$.

**Focal Loss**: As mentioned in §4, our proposed approach is the sample-dependent schedule FLSD-53 ($\gamma = 5$ for $\hat{p}_y \in [0, 0.2)$, and $\gamma = 3$ for $\hat{p}_y \in [0.2, 1]$), which we find to perform well across most classification datasets and network architectures. In addition, we also train other focal loss baselines, including ones with $\gamma$ fixed to 1, 2 and 3, and also ones that have a training epoch-dependent schedule for $\gamma$. Among the focal loss models trained with a fixed $\gamma$, using validation set we find $\gamma = 3$ (FL-3) to perform the best. Details of all these approaches can be found in Appendix F.

**Temperature Scaling:** In order to compute the optimal temperature, we use two different methods: (a) learning the temperature by minimising val set NLL, and (b) performing grid search over tempera-

| Dataset | Model | Cross-Entropy | Brier Loss | MMCE | LS-0.05 | FL-3 (Ours) | FLSD-53 (Ours) |
|---|---|---|---|---|---|---|---|
| CIFAR-100 | ResNet-50 | 23.3 | 23.39 | 23.2 | 23.43 | 22.75 | 23.22 |
| | ResNet-110 | 22.73 | 25.1 | 23.07 | 23.43 | 22.92 | 22.51 |
| | Wide-ResNet-26-10 | 20.7 | 20.59 | 20.73 | 21.19 | 19.69 | 20.11 |
| | DenseNet-121 | 24.52 | 23.75 | 24.0 | 24.05 | 23.25 | 22.67 |
| CIFAR-10 | ResNet-50 | 4.95 | 5.0 | 4.99 | 5.29 | 5.25 | 4.98 |
| | ResNet-110 | 4.89 | 5.48 | 5.4 | 5.52 | 5.08 | 5.42 |
| | Wide-ResNet-26-10 | 3.86 | 4.08 | 3.91 | 4.2 | 4.13 | 4.01 |
| | DenseNet-121 | 5.0 | 5.11 | 5.41 | 5.09 | 5.33 | 5.46 |
| Tiny-ImageNet | ResNet-50 | 49.81 | 53.2 | 51.31 | 47.12 | 49.69 | 49.06 |
| 20 Newsgroups | Global Pooling CNN | 26.68 | 27.06 | 27.23 | 26.03 | 29.26 | 27.98 |
| SST Binary | Tree-LSTM | 12.85 | 12.85 | 11.86 | 13.23 | 12.19 | 12.8 |

Table 2: Test set error ($\%$) computed for different approaches.

| Dataset | Model | Cross-Entropy | | Brier Loss | | MMCE | | LS-0.05 | | FL-3 (Ours) | | FLSD-53 (Ours) | |
|---|---|---|---|---|---|---|---|---|---|---|---|---|---|
| | | Pre T | Post T | Pre T | Post T | Pre T | Post T | Pre T | Post T | Pre T | Post T | Pre T | Post T |
| CIFAR-10/SVHN | ResNet-110 | 61.71 | 59.66 | 94.80 | 95.13 | 85.31 | 85.39 | 68.68 | 68.68 | **96.74** | **96.92** | 90.83 | 90.97 |
| | Wide-ResNet-26-10 | 96.82 | 97.62 | 94.51 | 94.51 | 97.35 | 97.95 | 84.63 | 84.66 | 98.19 | 98.05 | **98.29** | **98.20** |
| CIFAR-10/CIFAR-10-C | ResNet-110 | 77.53 | 75.16 | 84.09 | 83.86 | 71.96 | 70.02 | 72.17 | 72.18 | 82.27 | 82.18 | **85.05** | **84.70** |
| | Wide-ResNet-26-10 | 81.06 | 80.68 | 85.03 | 85.03 | 82.17 | 81.72 | 71.10 | 71.16 | 82.17 | 81.86 | **87.05** | **87.30** |

Table 3: AUROC ($\%$) computed for models trained on CIFAR-10 (in-distribution), and using SVHN and CIFAR-10-C (Gaussian Noise corruption with severity level 5) respectively as the OoD datasets.

tures between 0 and 10, with a step of 0.1, and finding the one that minimises val set ECE. We find the second approach to produce *stronger baselines* and report results obtained using this approach.

**Performance Gains:** We report ECE$\%$ (computed using 15 bins) along with optimal temperatures in Table 1, and test set error in Table 2. We report the other calibration scores (AdaECE, Classwise-ECE, MCE and NLL) in Appendix F. Firstly, for all dataset-network pairs, we obtain very competitive classification accuracies (shown in Table 2). Secondly, *it is clear from Table 1, and Tables F.1 and F.2 in the appendix, that focal loss with sample-dependent $\gamma$ and with $\gamma = 3$ outperform all the baselines: cross-entropy, label smoothing, Brier loss and MMCE.* They broadly produce the lowest calibration errors *both before and after temperature scaling*. This observation is particularly encouraging, as it also indicates that a principled method of obtaining values of $\gamma$ for focal loss can produce a very calibrated model, with no need to use validation set for tuning $\gamma$. As shown in Figure 4, we also compute $90\%$ confidence intervals for ECE, AdaECE and Classwise-ECE using 1000 bootstrap samples following [15], and using ResNet-50/110 trained on CIFAR-10 (see Appendix G for more results). Note that FLSD-53 produces the lowest calibration errors in general, and the difference in the metric values between FLSD-53 and other approaches (except Brier loss) is mostly statistically significant (i.e., confidence intervals don't overlap), especially before temperature scaling. In addition to the lower calibration errors, there are other advantages of focal loss as well, which we explore next.

**More advantages of focal loss:** *Behaviour on Out-of-Distribution (OoD) data:* A perfectly calibrated model should have low confidence whenever it misclassifies, including when it encounters data which is OoD [34]. Although temperature scaling calibrates a model under the i.i.d. assumption, it is known to fail under distributional shift [27]. Since focal loss has implicit regularisation effects on the network (see §4), we investigate if it helps to learn representations that are more robust to OoD data. To do this, we use ResNet-110 and Wide-ResNet-26-10 trained on CIFAR-10 and consider the SVHN [23] test set and CIFAR-10-C [9] with Gaussian noise corruption at severity 5 as OoD data. We use the entropy of the softmax distribution as the measure of confidence or uncertainty, and report the corresponding AUROC scores both before and after temperature scaling in Table 3. For both SVHN and CIFAR-10-C (using Gaussian noise), models trained on focal loss clearly obtain the highest AUROC scores. *Note that Focal loss even without temperature scaling performs better than other methods with temperature scaling.* We also present the ROC plots pre and post temperature scaling for models trained on CIFAR-10 and tested on SVHN in Figure 5. Thus, it is quite encouraging to note that models trained on focal loss are not only better calibrated under the i.i.d. assumption, but also seem to perform better than other competitive loss functions when we try shifting the distribution from CIFAR-10 to SVHN or CIFAR-10-C (pre and post temperature scaling).

*Confident and Calibrated Models:* It is worth noting that focal loss with sample-dependent $\gamma$ has optimal temperatures that are very close to 1, mostly lying between 0.9 and 1.1 (see Table 1). This property is shown by the Brier loss and label smoothing models as well, albeit with worse calibration

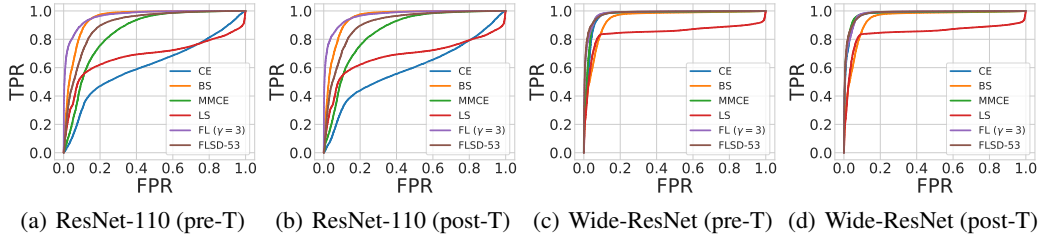

(a) ResNet-110 (pre-T)　　(b) ResNet-110 (post-T)　(c) Wide-ResNet (pre-T)　(d) Wide-ResNet (post-T)

Figure 5: ROC plots obtained from ResNet-110 and Wide-ResNet-26-10 architectures trained on CIFAR-10 (in-distribution) and tested on SVHN (OoD), both pre and post temperature scaling.

errors. By contrast, the temperatures for cross-entropy and MMCE models are significantly higher, with values lying between 2.0 and 2.8. An optimal temperature close to 1 indicates that the model is innately calibrated, and cannot be made significantly more calibrated by temperature scaling. In fact, a temperature much greater than 1 can make a model underconfident in general, as it is applied irrespective of the correctness of model outputs. We observe this empirically for ResNet-50 and ResNet-110 trained on CIFAR-10. Although models trained with cross-entropy have much higher confidence before temperature scaling than those trained with focal loss, after temperature scaling, focal loss models are significantly more confident in their predictions. We provide quantitative and qualitative empirical results to support this claim in Appendix H.

## 6    Conclusion

In this paper, we have studied the properties of focal loss, an alternative loss function that can yield classification networks that are more naturally calibrated than those trained using the conventional cross-entropy loss, while maintaining accuracy. In particular, we show in §4 that focal loss implicitly maximises entropy while minimising the KL divergence between the predicted and the target distributions. We also show that, because of its design, it naturally regularises the weights of a network during training, reducing NLL overfitting and thereby improving calibration. Furthermore, we empirically observe that models trained using focal loss are not only better calibrated under i.i.d. assumptions, but can also be better at detecting OoD samples which we show by taking CIFAR-10 as the in-distribution dataset and SVHN and CIFAR-10-C as out-of-distribution datasets, something which temperature scaling fails to achieve.

# 7 Broader Impact

Our work shows that using the right kind of loss function can lead to a calibrated model. This helps in improving the reliability of these models when used in real-world applications. It can help in deployment of the models such that users can be alerted when its prediction may not be trustworthy. We do not directly see a situation where calibrated neural networks can have a negative impact on society, but we do believe that research on making models more calibrated will help improve fairness and trust in AI.

## Acknowledgments and Disclosure of Funding

This work was started whilst J Mukhoti was at FiveAI, and completed after he moved to the University of Oxford. V Kulharia is wholly funded by a Toyota Research Institute grant. A Sanyal acknowledges support from The Alan Turing Institute under the Turing Doctoral Studentship grant TU/C/000023. This work was also supported by the Royal Academy of Engineering under the Research Chair and Senior Research Fellowships scheme, EPSRC/MURI grant EP/N019474/1 and FiveAI.

## Footnotes

[2]We note in passing that unlike cross-entropy loss, focal loss in its general form is not a proper loss function, as minimising it does not always lead to the predicted distribution $\hat{p}$ being equal to the target distribution $q$ (see Appendix B for the relevant definition and a longer discussion). However, when $q$ is a one-hot encoding (as in our case, and for most classification tasks), minimising focal loss does lead to $\hat{p}$ being equal to $q$.

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
