[Supplementary Material]

Figure A.1: The confidence values for training samples at different epochs during the NLL training of a ResNet-50 on CIFAR-10 (see §3). Top row: reliability plots using 25 confidence bins; bottom row: % of samples in each bin. As training progresses, the model gradually shifts all training samples to the highest confidence bin. Notably, it continues to do so even after achieving 100% training accuracy by the 300 epoch point.

## Appendix: Calibrating Deep Neural Networks using Focal Loss

In §A, we provide some empirical evidence for the observation made in §3 in the main paper using reliability plots. In §B, we discuss the relation between focal loss and a regularised KL divergence, where the regulariser is the entropy of the predicted distribution. In §C, we discuss the regularisation effect of focal loss on a simple setup, i.e. a generalised linear model trained on a simple data distribution. In §D, we show the proofs of the two propositions formulated in the main text. We then describe all the datasets and implementation details for our experiments in §E. In §F, we discuss additional approaches for training using focal loss, and also the results we get from these approaches. We also provide the Top-5 accuracies of several models as we speculate that calibrated models should have a higher softmax probability on the correct class even when they misclassify, as compared to models which are less calibrated. We further provide the results of evaluating our models using various metrics other than ECE (like AdaECE, Classwise-ECE, MCE and NLL). Next, in §G, we provide additional results related to the confidence interval experiments performed in §5 of the main paper. In §H, we provide empirical and qualitative results to show that models trained using focal loss are calibrated, whilst maintaining their confidence on correct predictions. In §I, we provide a brief extension of our discussion about Figure 2(e) in the main paper, with a plot of $L_2$ norms of features obtained from the last ResNet block during training. In §J, we provide some empirical evidence to support the claims we make in §4 of the main paper about early stopping. Finally, in §K, we discuss the performance of focal loss on the downstream task of machine translation with beam search. We choose machine translation as the downstream task because in machine translation, softmax vectors from a model are directly fed into the beam search algorithm, and hence more calibrated probability vectors should intuitively produce better translations

## A    Reliability Plots

In this section, we provide some empirical evidence to support the observation made in §3 of the main paper that a model, even after attaining perfect training accuracy, can reduce the training NLL (loss) further by increasing the prediction confidences to match the ground-truth one-hot encoding. To empirically observe this, we use the ResNet-50 network used for the analysis in §3. We divide the confidence range $[0, 1]$ into 25 bins, and present reliability plots computed on the training set at training epochs 100, 200, 300 and 350 (see the top row of Figure A.1). In Figure A.1, we also show the percentage of samples in each confidence bin. It is quite clear from these plots that over time, the network gradually pushes all of the training samples towards the highest confidence bin. Furthermore, even though the network has achieved 100% accuracy on the training set by epoch 300, it still pushes some of the samples lying in lower confidence bins to the highest confidence bin by epoch 350.

## B    Relation between Focal Loss and Entropy Regularised KL Divergence

Here we show why focal loss favours accurate but relatively less confident solutions. We show that it inherently provides a trade-off between minimizing the KL-divergence and maximizing the entropy, depending on the strength of $\gamma$. We use $\mathcal{L}_f$ and $\mathcal{L}_c$ to denote the focal loss with parameter $\gamma$ and cross entropy between $\hat{p}$ and $q$, respectively. $K$ denotes the number of classes and $q_y$ denotes the ground-truth probability assigned to the $y^{th}$ class (similarly for $\hat{p}_y$). We consider the following simple extension of focal loss:

$$
\begin{aligned}
\mathcal{L}_f &= -\sum_{y=1}^{K}(1-\hat{p}_y)^\gamma q_y \log \hat{p}_y \\
&\geq -\sum_{y=1}^{K}(1-\gamma\hat{p}_y)q_y \log \hat{p}_y && \text{By Bernoulli's inequality } \forall \gamma \geq 1, \text{ since } \hat{p}_y \in [0,1] \\
&= -\sum_{y=1}^{K} q_y \log \hat{p}_y - \gamma \left|\sum_{y=1}^{K} q_y \hat{p}_y \log \hat{p}_y\right| && \forall y, \log \hat{p}_y \leq 0 \\
&\geq -\sum_{y=1}^{K} q_y \log \hat{p}_y - \gamma \max_j q_j \sum_{y=1}^{K}|\hat{p}_y \log \hat{p}_y| && \text{By Hölder's inequality } ||fg||_1 \leq ||f||_\infty ||g||_1 \\
&\geq -\sum_{y=1}^{K} q_y \log \hat{p}_y + \gamma \sum_{y=1}^{K}\hat{p}_y \log \hat{p}_y && \forall j, q_j \in [0,1] \\
&= \mathcal{L}_c - \gamma \mathbb{H}[\hat{p}].
\end{aligned}
$$

We know that $\mathcal{L}_c = \mathrm{KL}(q||\hat{p}) + \mathbb{H}[q]$. Combining this equality with the above inequality leads to:

$$
\mathcal{L}_f \geq \mathrm{KL}(q||\hat{p}) + \underbrace{\mathbb{H}[q]}_{constant} - \gamma \mathbb{H}[\hat{p}].
$$

In the case of one-hot encoding (Delta distribution for $q$), focal loss will maximize $-\hat{p}_y \log \hat{p}_y$ (let $y$ be the ground-truth class index), the component of the entropy of $\hat{p}$ corresponding to the ground-truth index. Thus, it will prefer learning $\hat{p}$ such that $\hat{p}_y$ is assigned a higher value (because of the KL term), but not too high (because of the entropy term), and will ultimately avoid preferring overconfident models (by contrast to cross-entropy loss). Experimentally, we found the solution of the cross-entropy and focal loss equations, i.e. the value of the predicted probability $\hat{p}$ which minimises the loss, for various values of $q$ in a binary classification problem (i.e. $K = 2$), and plotted it in Figure B.1. As expected, focal loss favours a more entropic solution $\hat{p}$ that is closer to $0.5$. In other words, as Figure B.1 shows, solutions to focal loss (Equation 2) will always have higher entropy than those of cross-entropy, depending on the value of $\gamma$.

Figure B.1: Optimal $\hat{p}$ for various values of $q$.

$$
\hat{p} = \mathrm{argmin}_x \; -(1-x)^\gamma q \log x - x^\gamma (1-q) \log (1-x) \quad 0 \leq x \leq 1 \quad (2)
$$

## C    Focal Loss and Cross-Entropy on a Linear Model

The behaviour of deep neural networks is generally quite different from linear models and the problem of calibration is more pronounced in the case of deep neural networks, hence we focus on analysing the calibration of deep networks in the paper. However, weight norm analysis for the initial layers is complex due to batchnorm and weight decay. Hence, to see the effect of weight magnification on miscalibration, here we use a simple network without batchnorm or weight decay, which is a generalised linear model, and a simple data distribution.

**Setup**    We consider a binary classification problem. The data matrix $\mathbf{X} \in \mathbb{R}^{2 \times N}$ is created by assigning each class, two normally distributed clusters such that the mean of the clusters are linearly

Figure C.1: (a): Norm of logits (b): Norm of weights.

separable. The mean of the clusters are situated on the vertices of a two-dimensional hypercube of side length 4. The standard deviation for each cluster is 1 and the samples are randomly linearly combined within each cluster in order to add covariance. Further, for $10\%$ of the data points, the labels were flipped. 4000 samples are used for training and 1000 samples are used for testing. The model consists of a simple 2-parameter logistic regression model. $f_{\mathbf{w}}(\mathbf{x}) = \sigma(w_1 x_1 + w_2 x_2)$. We train this model using both cross-entropy and focal loss with $\gamma = 1$.

**Weight Magnification** We have argued that focal loss implicitly regularizes the weights of the model by providing smaller gradients as compared to cross-entropy. This helps in calibration as, if all the weights are large, the logits are large and thus the confidence of the network is large on all test points, even on the misclassified points. When the model misclassifies, it misclassifies with a high confidence. Figure C.1 shows, for a generalised linear model, that the norm of the logits and the weights of a network blows for Cross Entropy as compared to Focal Loss.

**High Confidence for mistakes** Figures C.2 (b) and (c) show that running gradient descent with cross-entropy (CE) and focal loss (FL) both gives the same decision regions i.e. the weight vector points in the same region for both FL and CE. However, as we have seen that the norm of the weights is much larger for CE as compared to FL, we would expect the confidence of misclassified test points to be large for CE as compared to FL. Figure C.2 (a) plots a histogram of the confidence of the misclassified points and it shows that CE misclassifies almost always with greater than $90\%$ confidence whereas FL misclassifies with much lower confidence.

Figure C.2: (a): Confidence of mis-classifications (b): Decision boundary of linear classifier trained using cross entropy (c): Decision boundary of linear classifier trained using focal loss

# D   Proofs

Here we provide the proofs of both the propositions presented in the main text. While Proposition 1 helps us understand the regularization effect of focal loss, Proposition 2 provides us the $\gamma$ values in a principled way such that it is sample-dependent. Implementing the sample-dependent $\gamma$ is very easy as implementation of the Lambert-W function [4] is available in standard libraries (e.g. python scipy).

**Proposition 1.** *For focal loss $\mathcal{L}_f$ and cross-entropy $\mathcal{L}_c$, the gradients $\frac{\partial \mathcal{L}_f}{\partial \mathbf{w}} = \frac{\partial \mathcal{L}_c}{\partial \mathbf{w}} g(\hat{p}_{i,y_i}, \gamma)$, where $g(p, \gamma) = (1-p)^\gamma - \gamma p (1-p)^{\gamma-1} \log(p)$, $\gamma \in \mathbb{R}^+$ is the focal loss hyperparameter, and $\mathbf{w}$ denotes the parameters of the last linear layer. Thus $\left\| \frac{\partial \mathcal{L}_f}{\partial \mathbf{w}} \right\| \leq \left\| \frac{\partial \mathcal{L}_c}{\partial \mathbf{w}} \right\|$ if $g(\hat{p}_{i,y_i}, \gamma) \in [0, 1]$.*

*Proof.* Let $\mathbf{w}$ be the linear layer parameters connecting the feature map to the logit $s$. Then, using the chain rule, $\frac{\partial \mathcal{L}_f}{\partial \mathbf{w}} = \left( \frac{\partial s}{\partial \mathbf{w}} \right) \left( \frac{\partial \hat{p}_{i,y_i}}{\partial s} \right) \left( \frac{\partial \mathcal{L}_f}{\partial \hat{p}_{i,y_i}} \right)$. Similarly, $\frac{\partial \mathcal{L}_c}{\partial \mathbf{w}} = \left( \frac{\partial s}{\partial \mathbf{w}} \right) \left( \frac{\partial \hat{p}_{i,y_i}}{\partial s} \right) \left( \frac{\partial \mathcal{L}_c}{\partial \hat{p}_{i,y_i}} \right)$. The derivative of the focal loss with respect to $\hat{p}_{i,y_i}$, the softmax output of the network for the true class $y_i$, takes the form

$$\frac{\partial \mathcal{L}_f}{\partial \hat{p}_{i,y_i}} = -\frac{1}{\hat{p}_{i,y_i}} \Big( (1 - \hat{p}_{i,y_i})^\gamma - \gamma \hat{p}_{i,y_i} (1 - \hat{p}_{i,y_i})^{\gamma-1} \log(\hat{p}_{i,y_i}) \Big)$$

$$= \frac{\partial \mathcal{L}_c}{\partial \hat{p}_{i,y_i}} g(\hat{p}_{i,y_i}, \gamma),$$

in which $g(\hat{p}_{i,y_i}, \gamma) = (1 - \hat{p}_{i,y_i})^\gamma - \gamma \hat{p}_{i,y_i} (1 - \hat{p}_{i,y_i})^{\gamma-1} \log(\hat{p}_{i,y_i})$ and $\frac{\partial \mathcal{L}_c}{\partial \hat{p}_{i,y_i}} = -\frac{1}{\hat{p}_{i,y_i}}$. It is thus straightforward to verify that if $g(\hat{p}_{i,y_i}, \gamma) \in [0, 1]$, then $\left\| \frac{\partial \mathcal{L}_f}{\partial \hat{p}_{i,y_i}} \right\| \leq \left\| \frac{\partial \mathcal{L}_c}{\partial \hat{p}_{i,y_i}} \right\|$, which itself implies that $\left\| \frac{\partial \mathcal{L}_f}{\partial \mathbf{w}} \right\| \leq \left\| \frac{\partial \mathcal{L}_c}{\partial \mathbf{w}} \right\|$. $\square$

**Proposition 2.** *Given a $p_0$, for $1 \geq p \geq p_0 > 0$, $g(p, \gamma) \leq 1$ for all $\gamma \geq \gamma^* = \frac{a}{b} + \frac{1}{\log a} W_{-1} \big( -\frac{a^{(1-a/b)}}{b} \log a \big)$, where $a = 1 - p_0$, $b = p_0 \log p_0$, and $W_{-1}$ is the Lambert-W function [4]. Moreover, for $p \geq p_0 > 0$ and $\gamma \geq \gamma^*$, the equality $g(p, \gamma) = 1$ holds only for $p = p_0$ and $\gamma = \gamma^*$.*

*Proof.* We derive the value of $\gamma > 0$ for which $g(p_0, \gamma) = 1$ for a given $p_0 \in [0, 1]$. From Proposition 4.1, we already know that

$$\frac{\partial \mathcal{L}_f}{\partial \hat{p}_{i,y_i}} = \frac{\partial \mathcal{L}_c}{\partial \hat{p}_{i,y_i}} g(\hat{p}_{i,y_i}, \gamma), \tag{3}$$

where $\mathcal{L}_f$ is focal loss, $\mathcal{L}_c$ is cross entropy loss, $\hat{p}_{i,y_i}$ is the probability assigned by the model to the ground-truth correct class for the $i^{th}$ sample, and

$$g(\hat{p}_{i,y_i}, \gamma) = (1 - \hat{p}_{i,y_i})^\gamma - \gamma \hat{p}_{i,y_i} (1 - \hat{p}_{i,y_i})^{\gamma-1} \log(\hat{p}_{i,y_i}). \tag{4}$$

For $p \in [0, 1]$, if we look at the function $g(p, \gamma)$, then we can clearly see that $g(p, \gamma) \to 1$ as $p \to 0$, and that $g(p, \gamma) = 0$ when $p = 1$. To observe the behaviour of $g(p, \gamma)$ for intermediate values of $p$, we first take its derivative with respect to $p$:

$$\frac{\partial g(p, \gamma)}{\partial p} = \gamma (1-p)^{\gamma-2} \big[ -2(1-p) - (1-p)\log p + (\gamma-1)p \log p \big] \tag{5}$$

In Equation 5, $\gamma(1-p)^{\gamma-2} > 0$ except when $p = 1$ (in which case $\gamma(1-p)^{\gamma-2} = 0$). Thus, to observe the sign of the gradient $\frac{\partial g(p, \gamma)}{\partial p}$, we focus on the term

$$-2(1-p) - (1-p)\log p + (\gamma-1)p \log p. \tag{6}$$

Dividing Equation 6 by $(-\log p)$, the sign remains unchanged and we get

$$k(p, \gamma) = \frac{2(1-p)}{\log p} + 1 - \gamma p. \tag{7}$$

We can see that $k(p, \gamma) \to 1$ as $p \to 0$ and $k(p, \gamma) \to -(1 + \gamma)$ as $p \to 1$ (using l'Hôpital's rule). Furthermore, $k(p, \gamma)$ is monotonically decreasing for $p \in [0, 1]$. Thus, as the gradient $\frac{\partial g(p, \gamma)}{\partial p}$ is positive initially starting from $p = 0$ and negative later till $p = 1$, we can say that $g(p, \gamma)$ first monotonically increases starting from 1 (as $p \to 0$) and then monotonically decreases down to 0 (at $p = 1$). Thus, if for some threshold $p_0 > 0$ and for some $\gamma > 0$, $g(p, \gamma) = 1$, then $\forall p > p_0$,

$g(p, \gamma) < 1$. We now want to find a $\gamma$ such that $\forall p \geq p_0$, $g(p, \gamma) \leq 1$. First, let $a = (1 - p_0)$ and $b = p_0 \log p_0$. Then:

$$
\begin{aligned}
g(p_0, \gamma) &= (1 - p_0)^{\gamma} - \gamma p_0 (1 - p_0)^{\gamma - 1} \log p_0 \leq 1 \\
&\implies (1 - p_0)^{\gamma - 1}[(1 - p_0) - \gamma p_0 \log p_0] \leq 1 \\
&\implies a^{\gamma - 1}(a - \gamma b) \leq 1 \\
&\implies (\gamma - 1) \log a + \log(a - \gamma b) \leq 0 \\
&\implies \left(\gamma - \frac{a}{b}\right) \log a + \log(a - \gamma b) \leq \left(1 - \frac{a}{b}\right) \log a \\
&\implies (a - \gamma b) e^{(\gamma - a/b) \log a} \leq a^{(1 - a/b)} \\
&\implies \left(\gamma - \frac{a}{b}\right) e^{(\gamma - a/b) \log a} \leq -\frac{a^{(1 - a/b)}}{b} \\
&\implies \left(\left(\gamma - \frac{a}{b}\right) \log a\right) e^{(\gamma - a/b) \log a} \geq -\frac{a^{(1 - a/b)}}{b} \log a
\end{aligned}
\tag{8}
$$

where $a = (1 - p_0)$ and $b = p_0 \log p_0$. We know that the inverse of $y = xe^x$ is defined as $x = W(y)$, where $W$ is the Lambert-W function [4]. Furthermore, the r.h.s. of the inequality in Equation 8 is always negative, with a minimum possible value of $-1/e$ that occurs at $p_0 = 0.5$. Therefore, applying the Lambert-W function to the r.h.s. will yield two real solutions (corresponding to a principal branch denoted by $W_0$ and a negative branch denoted by $W_{-1}$). We first consider the solution corresponding to the negative branch (which is the smaller of the two solutions):

$$
\begin{aligned}
\left((\gamma - \frac{a}{b}) \log a\right) &\leq W_{-1}\left(-\frac{a^{(1 - a/b)}}{b} \log a\right) \\
\implies \gamma &\geq \frac{a}{b} + \frac{1}{\log a} W_{-1}\left(-\frac{a^{(1 - a/b)}}{b} \log a\right)
\end{aligned}
\tag{9}
$$

If we consider the principal branch, the solution is

$$
\gamma \leq \frac{a}{b} + \frac{1}{\log a} W_0\left(-\frac{a^{(1 - a/b)}}{b} \log a\right),
\tag{10}
$$

which yields a negative value for $\gamma$ that we discard. Thus Equation 9 gives the values of $\gamma$ for which if $p > p_0$, then $g(p, \gamma) < 1$. In other words, $g(p_0, \gamma) = 1$, and for any $p < p_0$, $g(p, \gamma) > 1$. $\quad\square$

## E  Dataset Description and Implementation Details

We use the following image and document classification datasets in our experiments:

1. **CIFAR-10** [13]: This dataset has 60,000 colour images of size $32 \times 32$, divided equally into 10 classes. We use a train/validation/test split of 45,000/5,000/10,000 images.

2. **CIFAR-100** [13]: This dataset has 60,000 colour images of size $32 \times 32$, divided equally into 100 classes. (Note that the images in this dataset are not the same images as in CIFAR-10.) We again use a train/validation/test split of 45,000/5,000/10,000 images.

3. **Tiny-ImageNet** [6]: Tiny-ImageNet is a subset of ImageNet with 64 x 64 dimensional images, 200 classes and 500 images per class in the training set and 50 images per class in the validation set. The image dimensions of Tiny-ImageNet are twice that of CIFAR-10/100 images.

4. **20 Newsgroups** [17]: This dataset contains 20,000 news articles, categorised evenly into 20 different newsgroups based on their content. It is a popular dataset for text classification. Whilst some of the newsgroups are very related (e.g. rec.motorcycles and rec.autos), others are quite unrelated (e.g. sci.space and misc.forsale). We use a train/validation/test split of 15,098/900/3,999 documents.

5. **Stanford Sentiment Treebank (SST)** [32]: This dataset contains movie reviews in the form of sentence parse trees, where each node is annotated by sentiment. We use the dataset version with binary labels, for which 6,920/872/1,821 documents are used as the training/validation/test split. In the training set, each node of a parse tree is annotated as positive, neutral or negative. At test time, the evaluation is done based on the model classification at the root node, i.e. considering the whole sentence, which contains only positive or negative sentiment.

All our experiments required a single 12 GB TITAN Xp GPU. For training networks on CIFAR-10 and CIFAR-100, we use SGD with a momentum of 0.9 as our optimiser, and train the networks for 350 epochs, with a learning rate of 0.1 for the first 150 epochs, 0.01 for the next 100 epochs, and 0.001 for the last 100 epochs. We use a training batch size of 128. Furthermore, we augment the training images by applying random crops and random horizontal flips. For Tiny-ImageNet, we train for 100 epochs with a learning rate of 0.1 for the first 40 epochs, 0.01 for the next 20 epochs and 0.001 for the last 40 epochs. We use a training batch size of 64. It should be noted that for Tiny-ImageNet, we saved 50 samples per class (i.e., a total of 10000 samples) from the training set as our own validation set to fine-tune the temperature parameter (hence, we trained on 90000 images) and we use the Tiny-ImageNet validation set as our test set.

For 20 Newsgroups, we train the Global Pooling Convolutional Network [18] using the Adam optimiser, with learning rate $0.001$, and betas $0.9$ and $0.999$. The code is a PyTorch adaptation of [24]. We used Glove word embeddings [28] to train the network. We trained all the models for 50 epochs and used the models with the best validation accuracy.

For the SST Binary dataset, we train the Tree-LSTM [33] using the AdaGrad optimiser with a learning rate of $0.05$ and a weight decay of $10^{-4}$, as suggested by the authors. We used the constituency model, which considers binary parse trees of the data and trains a binary Tree-LSTM on them. The Glove word embeddings [28] were also tuned for best results. The code framework we used is inspired by [35]. We trained these models for 25 epochs and used the models with the best validation accuracy.

For all our models, we use the PyTorch framework, setting any hyperparameters not explicitly mentioned to the default values used in the standard models. For MMCE, we used $\lambda = 2$ for all our experiments as we found it to perform better over all the values we tried. A calibrated model which does not generalise well to an unseen test set is not very useful. Hence, for all the experiments, we set the training parameters in a way such that we get best test set accuracies on all datasets for each model.

## F    Additional Results

In addition to the sample-dependent $\gamma$ approach, we try the following focal loss approaches as well:

**Focal Loss (Fixed $\gamma$)**: We trained models on focal loss with $\gamma$ fixed to $1, 2$ and $3$. We found $\gamma = 3$ to produce the best ECE among models trained using a fixed $\gamma$. This corroborates the observation we made in §4 of the main paper that $\gamma = 3$ should produce better results than $\gamma = 1$ or $\gamma = 2$, as the regularising effect for $\gamma = 3$ is higher.

**Focal Loss (Scheduled $\gamma$)**: As a simplification to the sample-dependent $\gamma$ approach, we also tried using a schedule for $\gamma$ during training, as we expect the value of $\hat{p}_{i,y_i}$ to increase in general for all samples over time. In particular, we report results for two different schedules: (a) Focal Loss (scheduled $\gamma$ 5,3,2): $\gamma = 5$ for the first 100 epochs, $\gamma = 3$ for the next 150 epochs, and $\gamma = 2$ for the last 100 epochs, and (b) Focal Loss (scheduled $\gamma$ 5,3,1): $\gamma = 5$ for the first 100 epochs, $\gamma = 3$ for the next 150 epochs, and $\gamma = 1$ for the last 100 epochs. We also tried various other schedules, but found these two to produce the best results on the validation sets.

Finally, for the sample-dependent $\gamma$ approach, we also found the policy: Focal Loss (sample-dependent $\gamma$ 5,3,2) with $\gamma = 5$ for $\hat{p}_{i,y_i} \in [0, 0.2)$, $\gamma = 3$ for $\hat{p}_{i,y_i} \in [0.2, 0.5)$ and $\gamma = 2$ for $\hat{p}_{i,y_i} \in [0.5, 1]$ to produce competitive results.

In Tables F.1 and F.2, we present the AdaECE and Classwise-ECE scores for all the baselines discussed in Table 1 of the main paper.

In Table 2 of the main paper, we present the classification errors on the test datasets for all the major loss functions we considered. Here we also report the classification errors for the different focal loss approaches in Table F.6. We also report the ECE, Ada-ECE and Classwise-ECE for all the focal loss approaches in Table F.3, Table F.4 and Table F.5 respectively.

Finally, calibrated models should have a higher logit score (or softmax probability) on the correct class even when they misclassify, as compared to models which are less calibrated. Thus, intuitively, such models should have a higher Top-5 accuracy. In Table F.7, we report the Top-5 accuracies for all our models on datasets where the number of classes is relatively high (i.e., on CIFAR-100 with 100 classes and Tiny-ImageNet with 200 classes). We observe focal loss with sample-dependent $\gamma$ to

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

produce the highest top-5 accuracies on all models trained on CIFAR-100 and the second best top-5 accuracy (only marginally below the highest accuracy) on Tiny-ImageNet.

In addition to ECE, Ada-ECE and Classwise-ECE, we use various other metrics to compare the proposed methods with the baselines (i.e. cross-entropy, Brier loss, MMCE and Label Smoothing). We present the test NLL % before and after temperature scaling in Tables F.8 and F.9, respectively. We report the test set MCE % before and after temperature scaling in Tables F.10 and F.11, respectively.

We use the following abbreviation to report results on different varieties of Focal Loss. FL-1 refers to Focal Loss (fixed $\gamma$ 1), FL-2 refers to Focal Loss (fixed $\gamma$ 2), FL-3 refers to Focal Loss (fixed $\gamma$ 3), FLSc-531 refers to Focal Loss (scheduled $\gamma$ 5,3,1), FLSc-532 refers to Focal Loss (scheduled $\gamma$ 5,3,2), FLSD-532 refers to Focal Loss (sample-dependent $\gamma$ 5,3,2) and FLSD-53 refers to Focal Loss (sample-dependent $\gamma$ 5,3).

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

Table F.6: Error (%) computed for different focal loss approaches.

| Dataset | Model | Cross-Entropy | | Brier Loss | | MMCE | | LS-0.05 | | FLSD-53 (Ours) | |
|---|---|---|---|---|---|---|---|---|---|---|---|
| | | Top-1 | Top-5 | Top-1 | Top-5 | Top-1 | Top-5 | Top-1 | Top-5 | Top-1 | Top-5 |
| CIFAR-100 | ResNet-50 | 76.7 | 93.77 | 76.61 | 93.24 | 76.8 | 93.69 | 76.57 | 92.86 | 76.78 | **94.44** |
| | ResNet-110 | 77.27 | 93.79 | 74.9 | 92.44 | 76.93 | 93.78 | 76.57 | 92.27 | 77.49 | **94.78** |
| | Wide-ResNet-26-10 | 79.3 | 93.96 | 79.41 | 94.56 | 79.27 | 94.11 | 78.81 | 93.18 | 79.89 | **95.2** |
| | DenseNet-121 | 75.48 | 91.33 | 76.25 | 92.76 | 76 | 91.96 | 75.95 | 89.51 | 77.33 | **94.49** |
| Tiny-ImageNet | ResNet-50 | 50.19 | 74.24 | 46.8 | 70.34 | 48.69 | 73.52 | 52.88 | **76.15** | 50.94 | 76.07 |

Table F.7: Top-1 and Top-5 accuracies computed for different approaches.

| Dataset | Model | Cross-Entropy | Brier Loss | MMCE | LS-0.05 | FL-1 | FL-2 | FL-3 | FLSc-531 | FLSc-532 | FLSD-532 | FLSD-53 |
|---|---|---|---|---|---|---|---|---|---|---|---|---|
| CIFAR-100 | ResNet-50 | 153.67 | 99.63 | 125.28 | 121.02 | 105.61 | 92.82 | 87.52 | 100.09 | 92.66 | 94.1 | 88.03 |
| | ResNet-110 | 179.21 | 110.72 | 180.54 | 133.11 | 114.18 | 96.74 | 90.9 | 112.46 | 95.85 | 97.97 | 89.92 |
| | Wide-ResNet-26-10 | 140.1 | 84.62 | 119.58 | 108.06 | 87.56 | 77.8 | 74.66 | 88.61 | 78.52 | 78.86 | 76.92 |
| | DenseNet-121 | 205.61 | 98.31 | 166.65 | 142.04 | 115.5 | 93.11 | 87.13 | 107.91 | 93.12 | 91.14 | 85.47 |
| CIFAR-10 | ResNet-50 | 41.21 | 18.67 | 44.83 | 27.68 | 22.67 | 18.6 | 18.43 | 25.32 | 20.5 | 18.69 | 17.55 |
| | ResNet-110 | 47.51 | 20.44 | 55.71 | 29.88 | 22.54 | 19.19 | 17.8 | 32.77 | 22.48 | 19.39 | 18.54 |
| | Wide-ResNet-26-10 | 26.75 | 15.85 | 28.47 | 21.71 | 17.66 | 14.96 | 15.2 | 18.5 | 15.57 | 14.78 | 14.55 |
| | DenseNet-121 | 42.93 | 19.11 | 52.14 | 28.7 | 22.5 | 17.56 | 18.02 | 27.41 | 19.5 | 20.14 | 18.39 |
| Tiny-ImageNet | ResNet-50 | 232.85 | 240.32 | 234.29 | 235.04 | 219.07 | 202.92 | 207.2 | 217.52 | 211.42 | 204.71 | 204.97 |
| 20 Newsgroups | Global Pooling CNN | 176.57 | 130.41 | 158.7 | 90.95 | 140.4 | 115.97 | 109.62 | 128.75 | 123.72 | 124.03 | 109.17 |
| SST Binary | Tree-LSTM | 50.2 | 54.96 | 37.28 | 44.34 | 53.9 | 47.72 | 50.29 | 50.25 | 53.13 | 45.08 | 49.23 |

Table F.8: NLL (%) computed for different approaches pre temperature scaling.

| Dataset | Model | Cross-Entropy | Brier Loss | MMCE | LS-0.05 | FL-1 | FL-2 | FL-3 | FLSc-531 | FLSc-532 | FLSD-532 | FLSD-53 |
|---|---|---|---|---|---|---|---|---|---|---|---|---|
| CIFAR-100 | ResNet-50 | 106.83 | 99.57 | 101.92 | 120.19 | 94.58 | 91.80 | 87.37 | 92.77 | 91.58 | 92.83 | 88.27 |
| | ResNet-110 | 104.63 | 111.81 | 106.73 | 129.76 | 94.65 | 91.24 | 89.92 | 93.73 | 91.30 | 92.29 | 88.93 |
| | Wide-ResNet-26-10 | 97.10 | 85.77 | 95.92 | 108.06 | 83.68 | 80.44 | 74.66 | 84.11 | 80.01 | 80.40 | 78.14 |
| | DenseNet-121 | 119.23 | 98.74 | 113.24 | 136.28 | 100.81 | 91.35 | 87.55 | 98.16 | 91.55 | 90.57 | 86.06 |
| CIFAR-10 | ResNet-50 | 20.38 | 18.36 | 21.58 | 27.69 | 17.56 | 17.67 | 18.34 | 19.93 | 19.25 | 17.28 | 17.37 |
| | ResNet-110 | 21.52 | 19.60 | 24.61 | 29.88 | 17.32 | 17.53 | 17.62 | 23.79 | 20.21 | 17.78 | 18.24 |
| | Wide-ResNet-26-10 | 15.33 | 15.85 | 16.16 | 21.19 | 15.48 | 14.85 | 15.06 | 15.81 | 15.38 | 14.69 | 14.23 |
| | DenseNet-121 | 21.77 | 19.11 | 24.88 | 28.95 | 18.71 | 17.21 | 18.10 | 21.65 | 19.04 | 19.27 | 18.39 |
| Tiny-ImageNet | ResNet-50 | 220.98 | 238.98 | 226.29 | 214.95 | 217.51 | 202.92 | 207.20 | 215.37 | 211.57 | 205.42 | 204.97 |
| 20 Newsgroups | Global Pooling CNN | 87.95 | 93.11 | 99.74 | 90.42 | 87.24 | 93.60 | 94.69 | 97.89 | 93.66 | 91.73 | 93.98 |
| SST Binary | Tree-LSTM | 41.05 | 38.27 | 36.37 | 43.45 | 45.67 | 47.72 | 45.96 | 45.82 | 54.52 | 45.36 | 49.69 |

Table F.9: NLL (%) computed for different approaches post temperature scaling (cross-validating T on ECE).

| Dataset | Model | Cross-Entropy | Brier Loss | MMCE | LS-0.05 | FL-1 | FL-2 | FL-3 | FLSc-531 | FLSc-532 | FLSD-532 | FLSD-53 |
|---|---|---|---|---|---|---|---|---|---|---|---|---|
| CIFAR-100 | ResNet-50 | 44.34 | 36.75 | 39.53 | 26.11 | 33.22 | 21.03 | 13.02 | 26.76 | 23.56 | 22.4 | 16.12 |
| | ResNet-110 | 55.92 | 24.85 | 50.69 | 36.23 | 40.49 | 32.57 | 26 | 37.24 | 29.56 | 34.73 | 22.57 |
| | Wide-ResNet-26-10 | 49.36 | 14.68 | 40.13 | 23.79 | 27 | 15.14 | 9.96 | 27.81 | 17.59 | 13.64 | 10.17 |
| | DenseNet-121 | 56.28 | 15.47 | 49.97 | 43.59 | 35.45 | 21.7 | 11.61 | 38.68 | 18.91 | 21.34 | 9.68 |
| CIFAR-10 | ResNet-50 | 38.65 | 31.54 | 60.06 | 35.61 | 31.75 | 25 | 21.83 | 30.54 | 23.57 | 25.45 | 14.89 |
| | ResNet-110 | 44.25 | 25.18 | 67.52 | 45.72 | 73.35 | 25.92 | 25.15 | 34.18 | 30.38 | 30.8 | 18.95 |
| | Wide-ResNet-26-10 | 48.17 | 77.15 | 36.82 | 24.89 | 29.17 | 30.17 | 23.86 | 37.57 | 30.65 | 18.51 | 74.07 |
| | DenseNet-121 | 45.19 | 19.39 | 43.92 | 45.5 | 38.03 | 29.59 | 77.08 | 33.5 | 16.47 | 17.85 | 13.36 |
| Tiny-ImageNet | ResNet-50 | 30.83 | 8.41 | 26.48 | 25.48 | 20.7 | 8.47 | 6.11 | 16.03 | 9.28 | 8.97 | 3.76 |
| 20 Newsgroups | Global Pooling CNN | 36.91 | 31.35 | 34.72 | 8.93 | 34.28 | 24.1 | 18.85 | 26.02 | 25.02 | 24.29 | 17.44 |
| SST Binary | Tree-LSTM | 71.08 | 92.62 | 68.43 | 39.39 | 95.48 | 86.21 | 22.32 | 76.28 | 86.93 | 80.85 | 73.7 |

Table F.10: MCE (%) computed for different approaches pre temperature scaling.

| Dataset | Model | Cross-Entropy | Brier Loss | MMCE | LS-0.05 | FL-1 | FL-2 | FL-3 | FLSc-531 | FLSc-532 | FLSD-532 | FLSD-53 |
|---|---|---|---|---|---|---|---|---|---|---|---|---|
| CIFAR-100 | ResNet-50 | 12.75 | 21.61 | 11.99 | 18.58 | 8.92 | 8.86 | 6.76 | 7.46 | 6.76 | 5.24 | 27.18 |
| | ResNet-110 | 22.65 | 13.56 | 19.23 | 30.46 | 20.13 | 12 | 13.06 | 18.28 | 13.72 | 15.89 | 10.94 |
| | Wide-ResNet-26-10 | 14.18 | 13.42 | 16.5 | 23.79 | 10.28 | 18.32 | 9.96 | 13.18 | 11.01 | 12.5 | 9.73 |
| | DenseNet-121 | 21.63 | 8.55 | 13.02 | 29.95 | 10.49 | 11.63 | 6.17 | 6.21 | 6.48 | 9.41 | 5.68 |
| CIFAR-10 | ResNet-50 | 20.6 | 22.46 | 23.6 | 40.51 | 25.86 | 28.17 | 15.76 | 22.05 | 23.85 | 24.76 | 26.37 |
| | ResNet-110 | 29.98 | 22.73 | 31.87 | 45.72 | 29.74 | 23.82 | 37.61 | 26.25 | 25.94 | 11.59 | 17.35 |
| | Wide-ResNet-26-10 | 26.63 | 77.15 | 32.33 | 37.53 | 74.58 | 29.58 | 25.64 | 28.63 | 20.23 | 19.68 | 36.56 |
| | DenseNet-121 | 32.52 | 19.39 | 27.03 | 53.57 | 19.68 | 22.71 | 76.27 | 21.05 | 32.76 | 35.06 | 13.36 |
| Tiny-ImageNet | ResNet-50 | 13.33 | 12.82 | 12.52 | 17.2 | 6.5 | 8.47 | 6.11 | 5.97 | 7.01 | 5.73 | 3.76 |
| 20 Newsgroups | Global Pooling CNN | 36.91 | 31.35 | 34.72 | 8.93 | 34.28 | 24.1 | 18.85 | 26.02 | 25.02 | 24.29 | 17.44 |
| SST Binary | Tree-LSTM | 88.48 | 91.86 | 32.92 | 35.72 | 87.77 | 86.21 | 74.52 | 54.27 | 88.85 | 82.42 | 76.71 |

Table F.11: MCE (%) computed for different approaches post temperature scaling (cross-validating T on ECE).

# G   Bar plots

In this section, we present additional results in reference to Figure 4 in the main paper. In particular, we compute 90% confidence intervals for ECE, AdaECE and Classwise-ECE using 1000 bootstrap samples following [15] and present the resulting confidence intervals as bar plots in Figures G.1, G.2, G.3 and G.4. These plots further corroboate the observations made in Section 5 of the main paper. We find that FLSD-53 broadly produces the lowest calibration errors, and in quite a few cases

(a) ResNet-50 on CIFAR-100        (b) ResNet-110 on CIFAR-100

Figure G.1: Bar plots with confidence intervals for ECE, AdaECE and Classwise-ECE computed for ResNet-50 (first 3 figures) and ResNet-110 (last 3 figures) on CIFAR-100.

(a) Wide-ResNet-26-10 on CIFAR-100        (b) DenseNet-121 on CIFAR-100

Figure G.2: Bar plots with confidence intervals for ECE, AdaECE and Classwise-ECE computed for Wide-ResNet-26-10 (first 3 figures) and DenseNet-121 (last 3 figures) on CIFAR-100.

(a) Wide-ResNet-26-10 on CIFAR-10        (b) DenseNet-121 on CIFAR-10

Figure G.3: Bar plots with confidence intervals for ECE, AdaECE and Classwise-ECE computed for Wide-ResNet-26-10 (first 3 figures) and DenseNet-121 (last 3 figures) on CIFAR-10.

Figure G.4: Bar plots with confidence intervals for ECE, AdaECE and Classwise-ECE computed for ResNet-50 on Tiny-ImageNet.

(especially before temperature scaling) the differences in calibration errors between cross-entropy and focal loss are statistically significant.

# H    Focal Loss is Confident and Calibrated

In extension to what we present in Section 5 of the main paper, we also follow the approach adopted in [16], and measure the percentage of test samples that are predicted with a confidence of 0.99

| Dataset | Model | Cross-Entropy (Pre T) | | Cross-Entropy (Post T) | | MMCE (Pre T) | | MMCE (Post T) | | Focal Loss (Pre T) | | Focal Loss (Post T) | |
|---|---|---|---|---|---|---|---|---|---|---|---|---|---|
| | | $|S99|\%$ | Accuracy | $|S99|\%$ | Accuracy | $|S99|\%$ | Accuracy | $|S99|\%$ | Accuracy | $|S99|\%$ | Accuracy | $|S99|\%$ | Accuracy |
| CIFAR-10 | ResNet-110 | 97.11 | 96.33 | 11.5 | 97.39 | 97.65 | 96.72 | 10.62 | 99.83 | 61.41 | 99.51 | 31.10 | 99.68 |
| CIFAR-10 | ResNet-50 | 95.93 | 96.72 | 7.33 | 99.73 | 92.33 | 98.24 | 4.21 | 100 | 46.31 | 99.57 | 14.27 | 99.93 |

Table H.1: Percentage of test samples predicted with confidence higher than $99\%$ and the corresponding accuracy for Cross Entropy, MMCE and Focal loss computed both pre and post temperature scaling (represented in the table as pre T and post T respectively).

or more (we call this set of test samples $S99$). In Table H.1, we report $|S99|$ as a percentage of the total number of test samples, along with the accuracy of the samples in $S99$ for ResNet-50 and ResNet-110 trained on CIFAR-10, using cross-entropy loss, MMCE loss, and focal loss. We observe that $|S99|$ for the focal loss model is much lower than for the cross-entropy or MMCE models before temperature scaling. However, after temperature scaling, $|S99|$ for focal loss is significantly higher than for both MMCE and cross-entropy. The reason is that with an optimal temperature of 1.1, the confidence of the temperature-scaled model for focal loss does not reduce as much as those of the models for cross-entropy and MMCE, for which the optimal temperatures lie between 2.5 to 2.8. We thus conclude that models trained on focal loss are not only more calibrated, but also better preserve their confidence on predictions, even after being post-processed with temperature scaling.

Figure H.1: Qualitative results showing the performance of Cross Entropy, Brier loss, MMCE and Focal Loss (sample-dependent $\gamma$ 5,3) for a ResNet-50 trained on CIFAR-10. The first row of images have been correctly classified by networks trained on all four loss functions and the second row of images have all been incorrectly classified. For each image, we present the actual label, the predicted label and the confidence of the prediction both before and after temperature scaling.

In Figure H.1, we present some qualitative results to support this claim and show the improvement in the confidence estimates of focal loss in comparison to other baselines (i.e., cross entropy, MMCE and Brier loss). For this, we take ResNet-50 networks trained on CIFAR-10 using the four loss functions (cross entropy, MMCE, Brier loss and Focal loss with sample-dependent $\gamma$ 5,3) and measure the confidence of their predictions for four correctly and four incorrectly classified test samples. We report these confidences both before and after temperature scaling. It is clear from Figure H.1 that for all the correctly classified samples, the model trained using focal loss has very confident predictions both pre and post temperature scaling. However, on misclassified samples, we observe a very low confidence for the focal loss model. The ResNet-50 network trained using cross entropy is very confident even on the misclassified samples, particularly before temperature scaling. Apart from focal loss, the only model which has relatively low confidences on misclassified test samples is the one trained using Brier loss. These observations support our claim that focal loss produces not only a calibrated model but also one which is confident on its correct predictions.

## I   Ordering of Feature Norms

As an extension to the discussion related to Figure 2(e) in the main paper, we plot the $L_2$ norm of the features/activations obtained from the last ResNet block (right before the linear layer is applied on these features to get the logits). We plot these norms throughout the training period for networks trained on cross-entropy and focal loss with $\gamma$ set to $1, 2$ and $3$ in Figure I.1. We observe that there is a distinct ordering of feature norms for the four models: cross-entropy has the highest feature norm, followed by focal loss with $\gamma = 1$, followed by focal loss with $\gamma = 2$ and finally focal loss with $\gamma = 3$. Furthermore, this ordering is preserved throughout training. As we saw from Figure 2(e) in the main paper, from epoch 150 onwards (i.e., the epoch from which the networks start getting miscalibrated), there is a flip in the ordering of weight norms of the last linear layer. From epoch 150 onwards, the weight norms also follow the exact same ordering that we observe from Figure I.1 here. This shows that throughout training the initial layer weights (before last linear layer) of the network trained using focal loss are also regularized to favor lower norm of the output features, thus possibly leading to less peakiness in final prediction as compared to that of cross-entropy loss (see the 'Peak at the wrong place' paragraph of Section 3 of the main paper).

## J   Early stopping

From Figure 2(a), one may think that intermediate models (using early stopping) might provide better accuracy and calibration. However, there is no ideal approach for early stopping. For fair comparison, we train ResNet50, CIFAR-10 using cross-entrpy and focal loss with the best (in hindsight) possible early stopping. We train each model for 350 epochs and choose the 3 intermediate models with the best val set ECE, NLL and classification error, respectively. We present the test set performance in Table J.1.

Figure I.1: $L_2$ norm of features obtained from the last ResNet block (before the linear layer) of ResNet-50 averaged over entire training dataset of CIFAR-10 using a batch size of 128.

| Criterion | Loss | Epoch | Error | ECE % |
|-----------|---------|-------|-------|-------|
| ECE | CE | 151 | 7.34 | 1.69 |
| ECE | FLSD-53 | 257 | 5.52 | 0.85 |
| NLL | CE | 153 | 6.69 | 2.28 |
| NLL | FLSD-53 | 266 | 5.34 | 1.33 |
| Error | CE | 344 | 5.0 | 4.46 |
| Error | FLSD-53 | 343 | 4.99 | 1.43 |
| Full | CE | 350 | 4.95 | 4.35 |
| Full | FLSD-53 | 350 | 4.98 | 1.55 |

Table J.1: Classification errors and ECE scores obtained from ResNet-50 models trained using cross-entropy and focal loss with different early stopping criteria (best in hindsight ECE, NLL and classification error on the validation set) applied during training. In the table CE and FL stand for cross-entropy and focal loss respectively and the Full Criterion indicates models where early stopping has not been applied.

Figure K.1: Changes in *test set* BLEU score and *validation set* ECE with temperature, for models trained using (a) cross-entropy with hard targets (CE) (b) cross-entropy with label smoothing (LS) ($\alpha = 0.1$), and (c) focal loss (FL) ($\gamma = 1$).

From the table, we can observe that: 1) On every early stopping criterion, the model trained on focal loss outperforms the one trained on cross-entropy in both error and ECE, 2) ECE as a stopping criterion provides better test set ECE, but increases the test error significantly, 3) even without early stopping, focal loss achieves consistently better error and ECE compared to cross-entropy using any stopping criterion.

## K   Machine Translation: A Downstream Task

In this section, we explore machine translation with beam search as a relevant downstream task for calibration. Following the setup in [20], we train the Transformer architecture [37] on the WMT 2014 English-to-German translation dataset. The training settings (like optimiser, LR schedule, etc.) are the same as [37]. We chose machine translation as a relevant task because the softmax vectors produced by the transformer model are directly fed into the beam search algorithm, and hence softmax outputs from a calibrated model should intuitively produce better translations and a better BLEU score.

We train three transformer models, one on cross-entropy with hard target labels, the second on cross-entropy with label smoothing (with smoothing factor $\alpha = 0.1$) and the third on focal loss with $\gamma = 1$. In order to compare these models in terms of calibration, we report the test set ECE (%) both before and after temperature scaling in the first row of Table K.1. Furthermore, to evaluate their performance on the English-to-German translation task, we also report the test set BLEU score of these models in the second row of Table K.1. Finally, to study the variation of test set BLEU score and validation set ECE with temperature, we plot them against temperature for all three models in Figure K.1.

We observe from Table K.1 that the model trained on focal loss outperforms its competitors on ECE and also has a competitive edge over other methods on BLEU score as well. The focal loss

| Metrics | CE ($\alpha = 0.0$) | LS ($\alpha = 0.1$) | FL ($\gamma = 1.0$) |
|---------|-----------|-----------|-----------|
| ECE% Pre T / Post T / T | 10.16/2.59/1.2 | 3.25/3.25/1.0 | **1.69**/**1.69**/1.0 |
| BLEU Pre T / Post T | 26.31/26.21 | 26.33/26.33 | **26.39**/**26.39** |

Table K.1: Test set ECE and BLEU score both pre and post temperature scaling for cross-entropy (CE) with hard targets, cross-entropy with label smoothing (LS) ($\alpha = 0.1$) and focal loss (FL) ($\gamma = 1$).

model also has an optimal temperature of 1, just like the model trained on cross-entropy with label smoothing. From Figure K.1, we can see that the models obtain the highest BLEU scores at around the same temperatures at which they obtain low ECEs, thereby confirming our initial notion that a more calibrated model provides better translations. However, since the optimal temperatures are tuned on the validation set, they don't often correspond to the best BLEU scores on the test set. On the test set, the highest BLEU scores we observe are 26.33 for cross-entropy, 26.36 for cross-entropy with label smoothing, and 26.39 for focal loss. Thus, the performance of focal loss on machine translation (a downstream task related to calibration) is also very encouraging.