[Reviews · NeurIPS 2020]

Review 1

Summary and Contributions: This paper proposes a simple method for training calibrated deep neural networks. The paper evolves around an interesting finding of the authors regarding the focal loss (frequently used in computer vision for object detection and semantic segmentation to mitigate strong class imbalance by focusing on harder samples), which enforces better calibration. The authors show that focal loss minimizes the KL divergence between softmax predicted distribution and target distribution (similarly to NLL), and in addition increases the entropy of the predicted distribution and thus reducing overconfidence. Across the paper the authors argue the pertinence of the focal loss in the context of calibration of deep neural networks: they extend the study from Guo et al. [7] on the influence of the NLL on calibration and show that weight magnification is one of the main causes for miscalibration, they show formally the link between focal loss and NLL. Then the authors propose a hand-crafted policy for selecting the hyper-parameters of the focal loss of this use case and finally validate their findings over several datasets and architectures with convincing results. ---------------------- [Post-rebuttal update] The authors made a mostly convincing rebuttal and I will keep my rating and recommendation. I second the remarks from R2 and R5 for including vanilla Focal Loss results in the main paper and more importantly their comments regarding the OOD claims and results. If the authors claim that Focal Loss improves calibration and generalizes to OOD, the authors are expected to provide quantitative scores to support this: calibration scores (not yet provided) and AUROC (provided in the rebuttal). If the claims are not backed-up by quantitative results, please adjust the wording accordingly.

Strengths: - This paper is well written. The authors did a good job in arguing the utility of their method but also in illustrating the pathologies introduced by the NLL. I appreciate the detailed ablation studies that show the evolution during training of test NLL and entropy on correct and incorrect predictions, of calibration and various complementary metrics, weight norm. - The finding of the authors that the focal loss is an upper bound for the KL between predicted and target distributions with entropy regularization is interesting and novel. It allows connection to the practice of entropy regularization from other parts of machine learning. - The proposed method is simple and straightforward to implement, while achieving good calibration performance. An interesting perk of this method is that it's less dependent on the temperature scaling post-processing, i.e. the computed temperature is usually around 0.9-1.1 (similarly for Brier loss on images) - The authors present a large amount of convincing experiments across different datasets (CIFAR-10, CIFAR-100, Tiny Imagenet, 20 Newsgroups, SST), tasks (image classification, NLP) and architectures (ResNet-50, ResNet-110, WideResNet, DenseNet-121, Tree-LSTM)

Weaknesses: - It's not clear from the article if weight-decay was used for the experiments, on both the Cross Entropy and the Focal Loss. Weight-Decay has an non-negligeable effect on weight norms. The curves in the plot in Fig.2 e) would indicate the use of weight-decay but this is not mentioned in the text. Mind that some learning rate schedulers remove weight-decay for low learning rate values. Could the authors please clarify this aspect? - While the arguments of the authors make sense and are nicely described, the story would be more complete if the trend in weight magnification would be discussed beyond the last linear layer. Of course, the story-line risks being slightly different and more complex due to the BatchNorm layers that together with weight-decay lead to specific dynamics [i], adaptive learning rate when using weight decay. Recently, in [ii] authors show how to predict the evolution of the norms as a function of gradients norm and hyperparameters for BatchNorm networks. As said, the analysis would be more subtle, but I would be curious if the authors made their analysis beyond the last linear layer (which is typically outside the BatchNorm influence) Minor: - Focal Loss has been initially introduced to deal with strong class imbalance. In the current paper the loss with the proposed hyperparameter policy is used mostly on very well balanced datasets (CIFAR-10, CIFAR-100). While the calibration effect is obvious from the results obtained by the authors, what should we expect for imbalanced datasets: will it preserve the calibration for all classes? - The policy for selecting the hyperparameter \gamma for the focal loss feels hand-crafted, though it seems to do its job well. A related strategy in a similar spirit has been proposed in the Anchor Loss [iii] References: [i] Z. Li and S. Arora, An Exponential Learning Rate Schedule for Deep Learning, ICLR 2020 [ii] S. Roburin et al., Spherical Perspective on Learning with Batch Norm, arXiv 2020 [iii] Anchor Loss: Modulating Loss Scale based on Prediction Difficulty, ICCV 2019

Correctness: The paper is technically sound, I find both claims and method to be correct.

Clarity: Yes, the paper is clear.

Relation to Prior Work: The authors address well relation to prior works.

Reproducibility: Yes

Additional Feedback: Minor remark: - typo: L113: 0001 Suggestions: - here are few suggestions for improving this work and that could be potentially addressed in the rebuttal: + could the authors comment whether they did the authors run experiments with NLL + entropy regularization (eq. 1) instead of focal loss? If so, how well does it do? + could the authors please comment on the utilization of weight-decay and on the weight magnification phenomenon for other layers. + (minor) have the authors studied this work on imbalanced datasets? Conclusion: - This paper is nice and well written. The authors address a problem of interest, come up with a simple idea with good theoretical and empirical arguments. I would recommend this paper for acceptance.


Review 2

Summary and Contributions: This paper suggests using the focal loss to improve calibration. The focal loss emphasizes hard examples and it used in object detection, but this paper shows the focal loss (or rather, Focal Loss Sample-Dependent (5, 3)) can also be useful for calibration.

Strengths: There are very few real advances for improving calibration. Calibration is one of the most important aspects of ML reliability. The performance gains are marked.

Weaknesses: Minor: This paper shows many calibration measures. It shows the absolute deviation (l_1) and "maximum calibration error" (l_\infty), but it could show l_2 since that's the only l_p that is a proper scoring rule. (I know this isn't common due to Guo et al., but it Kumar NeurIPS 2019 show how to evaluate better.) Major: From the introduction: "Moreover, it is known that temperature scaling does not calibrate a model under data distribution shift [29]." "Finally, we also show that models trained on focal loss are significantly more robust to out-of-distribution samples as compared to models trained using cross-entropy, both before and after temperature scaling." This paper hardly shows performance under distribution shift. It briefly considers OOD detection not OOD generalization. Consequently it considers out-of-class examples, while usually calibration under distribution shows calibration on corrupted examples (Tiny-ImageNet-C). If they want to show that it is better for OOD detection, then show that the AUROC improves; histogram evaluation is meaningless without the AUROC. Likewise, they would need to show more than a comparison to SVHN since detecting OOD examples can vary wildly (inheriting the testbed from 1812.04606 could help). Alternatively and more simply, include evaluation on corrupted examples, like the work they cited: [29] or Snoek et al. Update: I was asking for calibration on CIFAR-10-C, not the detection of CIFAR-10-C. Consequently my main request was not fulfilled. Nonetheless I'll increase my score but please include these results.

Correctness: Yes

Clarity: This paper mixes its description of the sample dependent focal loss in a long text section with theory; separating these two would be useful since I'd prefer not to read about a "Lambert-W function" to find a concise description of the focal loss variant. AdaECE's description takes up some space in the paper but its results are hardly displayed in the main paper. Trimming this could make more room for results. Please show vanilla Focal Loss in the main table. _I worry that this paper created a more cumbersome and complex focal loss variant for perceived originality._

Relation to Prior Work: AdaECE should probably cite 1508.05154 who use adaptive binning (the only difference is that they use squared differences instead of absolute differences).

Reproducibility: Yes

Additional Feedback: I will easily and gladly change my score to 7 if authors _share adequate results on robustness to distribution shift_ and show performance of a vanilla focal loss in the main table since this that's simpler.


Review 3

Summary and Contributions: The authors describe a new approach to effectively calibrate neural networks using focal loss. The problem of miscalibrated neural networks particularly affects multiclass classification networks in that they make overconfident predictions. The described approach using focal loss results in well-calibrated networks while conserving good classification accuracies. The theoretical considerations are experimentally verified using various architectures for the classification of computer vision and NLP datasets. The work focuses on the following contributions: use of focal loss (with temperature scaling) to train models that are well calibrated; a thorough analysis of the factors causing miscalibration; proposal for a principled approach for focal loss hyperparameter tuning. ------ post-rebuttal update: Following the thorough and convincing rebuttal, I keep my pre-rebuttal rating. I also second R2's recommendations to adapt the language regarding the ODD claims and results.

Strengths: * The research problem is well described and set up. The authors demonstrate a good understanding of both the methods (and theoretical considerations behind them) and aspects that require improvements. * The proposed approach of using focal loss for network calibration is theoretically motivated and all claims are backed up through theoretical analysis and empirical findings on a number of architectures and datasets. Experimental results are both well described in the text and supported by figures that are accompanied by instructive captions. The superiority of focal loss for network calibration is demonstrated using appropriate baselines. Additional validation of the method on an odd dataset further corroborates its usefulness. * The proposed method/improvement constitutes a novel approach to mitigate network miscalibration without the sacrification of classification accuracy. While focal loss has gained more attention in the last year, this work highlights a new aspect with extensive theoretical considerations as well as empirical findings.

Weaknesses: The novelty might be limited but one might argue that this is outweighed by the importance of network miscalibration on the real-work deployment of classification networks. I missing a more in-depth discussion of why (or at least mention that) focal loss appears to work better on computer vision tasks as compared to the used text classification tasks.

Correctness: All theoretical claims are extensively backed up through correct profs. The empirical methodology is sufficient and correct.

Clarity: Overall, the paper is very well structured and well written. The logical structuring of the guides the reader through the thought process and sections build upon each other and are appropriately cross-referenced. The research question is well motivated and its relevance for the community worked out (though more close to life examples of the impact of miscalibrated models would be helpful) The necessary techniques and related work are introduced in a clear and understandable manner and methods and experimental settings are described in precise and clear language with critical findings appropriately highlighted. Some minor points: * Some sentences appear a little convoluted (e.g. page 1, 36-39, p2 line 51-54) * citation style in the related work part (around the end of page 1) makes it hard to read (printing out the names of the relevant authors would improve the reading experience). This also occurs in other parts of the manuscript. * some repetitions impair the reading experience a bit * The color coding (legend) in figure 2e is different from the other 4 plots in the figure. That’s confusing and should be corrected.

Relation to Prior Work: Yes, the authors describe how their work differs from previous attempts for the calibration of neural networks. Even though slightly different in their approach, a mention of ensembling/Bayesian approximation methods for network calibration would be to give more context.

Reproducibility: Yes

Additional Feedback:


Review 4

Summary and Contributions: The authors examine the use of focal loss (Lin et al. 2017) for obtaining well-calibrated deep neural networks. They begin by noticing that miscalibration arises in the training process due to overfitting of misclassified test examples (i.e. entropy in the predicted distributions approaches zero). They then show that focal loss can be interpreted as inducing (1) entropy regularization, as well as (2) weight regularization in the logit layer. Compared to cross-entropy loss, focal loss thus has the effect of alleviating overconfidence. Experiments are run on a variety of datasets (CIFAR-100, CIFAR-10, Tiny ImageNet, 20 Newsgroups, SST), showing improvements in ECE compared to cross-entropy loss. In these experiments the focal loss hyper-parameter gamma is chosen according to a proposed sample-dependent method that, intuitively, upweights cases where the predicted ground-truth class probability is less than 0.25 and downweights cases where it's greater than 0.25. --- Update: I thank the authors for their response. I am increasing my overall rating from 6 to a 7; accept. I am grateful to the authors for running additional experiments to include results in Table 1 for OOD detection. While this is an interesting result, I second R2's sentiment that the claims wrt OOD generalization as currently written are a bit too strong. I strongly suggest the authors update the wording in the paper's Introduction accordingly. Moreover, I'd like to second R2's suggestion of including results using vanilla Focal Loss in the main table. It's much simpler than the more cumbersome FLSD-53 and evidently achieves comparable results (Table F.10).

Strengths: The interpretation of focal loss as minimizing an entropy-regularized KL divergence between ground truth and predictions is, in my opinion, the strongest aspect of this work. I'd imagine this result to be of interest to the community, with implications for example in object detection where focal loss is already widely used. The analyses in Proposition 1 and in Figure 3(a) are particularly valuable. Additionally the training curves and entropies in Figure 1 and Figure 2 provide additional support for the prevailing understanding that overfitting is what causes miscalibration in neural networks. They provide supporting evidence for why regularization via weight decay, entropy regularization, and label smoothing tend to improve calibration. The paper is generally well-written and ties together prior works exploring these different regularization methods.

Weaknesses: The experimental results advocating for use of focal loss could use a bit of improvement. (1) Most significantly, I'm not yet convinced by the claim that focal loss is "significantly more robust to out-of-distribution data". While I do think it's promising that the predicted entropies for OOD data tend to be higher than for in-distribution data, it's difficult to conclude from the plots in Figure 5 alone that there is a signficant improvement. I'd insist the authors compare the AUROC/AUPRC that arise from using the softmax probabilities to detect OOD data, as in Hendrycks and Gimpel ICLR 2017. (Of course, the baselines of mixup and label smoothing need to be included as well.) Otherwise I'd err away from making the claim that focal loss yields an improvement. (2) An additional baseline -- in Table 1 I'd suggest the authors compare to mixup regularization (Thulasidasan et al. 2019) both with and without temperature scaling. (3) It's hypothesized (in the paragraph starting on line 197) that the networks may be magnifying logit layer weights to increase predicted confidences. But in Figure 2(e), the empirical weight norms are *decreasing* in all cases after epoch ~5. This is surprising since Figure 2(a) shows that overfitting doesn't occur until epoch ~150, so I would have expected weight norms to increase at least until then. Can the authors comment on this discrepancy? (4) What's behind the choice of threshold p = 0.25 in the proposed sample-dependent tuning method? Figure 3(a) shows that p = 0.25 is certainly a convenient choice that covers many gamma, but why 0.25 as opposed to say, 0.3? 0.5? What's the effect of tuning this p? It overall feels to me like trading off one hyperparameter (gamma) for several (p = 0.25, gamma = 3, gamma = 5). And to me at least, the empirical results in Appendix F suggest that FLSD-53 is not uniformly better than manually tuning gamma. Some additional suggestions: (5) I'd suggest the authors add confidence intervals to Table 1. (6) I'd suggest the authors add accuracy as a metric to Table 1. Appendix Table F.3 shows accuracies are more or less similar across the models, but it would be helpful to verify in the main text that the improvements in ECE are not arising at the expense of accuracy.

Correctness: Claims and methods look correct to me.

Clarity: The paper is generally well-written.

Relation to Prior Work: Yes, the paper does a good job reviewing prior work and the contribution is substantially different.

Reproducibility: Yes

Additional Feedback: Figure 1 and Figure 2 both show training curves. But in Figure 1 the *legend* indicates whether each subplot shows a train/test split, whereas in Figure 2 the *y axis* indicates whether each subplot shows a train/test split. This inconsistency is confusing; it'd be easier to interpret if this information was in the title of each subplot, for example. In Figure 4, I'd suggest the authors replace CIFAR-10 results with CIFAR-100 results, which is the dataset much more susceptible to miscalibration (Guo et al. 2017).

[Author Response · NeurIPS 2020]

| Dataset | Cross-Entropy | | LS-0.05 | | FLSD-53 (Ours) | | Cross-Entropy | | LS-0.05 | | FLSD-53 (Ours) | |
|---|---|---|---|---|---|---|---|---|---|---|---|---|
| | Pre T | Post T | Pre T | Post T | Pre T | Post T | Pre T | Post T | Pre T | Post T | Pre T | Post T |
| CIFAR-10/SVHN | 61.71 | 59.66 | 68.68 | 68.68 | **90.83** | **90.97** | 61.93 | 59.87 | 68.77 | 68.77 | **90.29** | **90.37** |
| CIFAR-10/CIFAR-10-C | 77.54 | 75.16 | 72.17 | 72.17 | **85.04** | **84.70** | 77.83 | 75.42 | 72.25 | 72.25 | **85.25** | **85.02** |

Table 1: AUROC (%) for out-of-distribution detection computed using ResNet-110 for CIFAR-10 and SVHN in the first row, and CIFAR-10 and CIFAR-10-Corrupted in the second row. In the left-hand table, we use softmax entropy to get the ROC curve. In the right-hand table, we use the confidence (maximum softmax probability) to get the ROC curve.

| Dataset | Cross-Entropy | | Brier Loss | | MMCE | | LS-0.05 | | FLSD-53 (Ours) | |
|---|---|---|---|---|---|---|---|---|---|---|
| | Pre T | Post T | Pre T | Post T | Pre T | Post T | Pre T | Post T | Pre T | Post T |
| CIFAR-10 | 5.62 | 3.79 (2.8) | 4.96 | 3.31 (1.2) | 6.37 | 5.15 (2.8) | 6.81 | 6.81 (1.0) | **4.14** | **3.30 (1.1)** |
| CIFAR-100 | 20.97 | 7.26 (2.3) | 11.75 | 5.09 (1.2) | 20.87 | 6.49 (2.3) | 14.65 | 10.00 (1.1) | **11.32** | **5.05 (1.2)** |

Table 2: L2 calibration error (%) computed for ResNet-110 on CIFAR-10 and CIFAR-100 (both pre and post temperature scaling). The optimal temperature is indicated in brackets. Best results have been marked in bold.

We thank the reviewers for their insightful comments. They found our proposed approach simple to implement (R1)
and practical (R3), our thorough analysis novel and valuable (R1,R3,R5) with good theoretical motivation (R1,R3), our
experiments exhaustive and our results convincing (R1,R2,R3,R5). We address their concerns below.

**OOD data detection (R2, R5)**: Thank you for suggesting the use of the AUROC metric. In Table 1, we present the
results of comparing our approach to CE and LS-0.05 (R5), using SVHN and CIFAR-10-C (corrupted using severity 5 of
the Gaussian noise, refer 1807.01697) (R5) separately as OOD data. We will report results from other corruption settings
in the appendix of the final version. We use softmax entropy (Fig. 5 in the main paper) and confidence (maximum
softmax probability) [32] respectively to compute the AUROC. **Note that FLSD-53 even without temperature scaling**
**performs better than other methods with temperature scaling.** Having said that, we would like to clarify here that
behavior on OOD data is not the main focus of the paper, and the primary purpose of this experiment is to show that (i)
temperature scaling may not work on OOD samples, and (ii) interestingly, models that are trained on focal loss can be
relatively more reliable. We will clarify this claim in the main paper.

$L_2$ **calibration error and other regularizers (R2, R5)**: We show the evaluation of ResNet-110 using RMS calibration
error [14] in Table 2 (R2). We will include numbers for other models and datasets in the main paper. We are aware
of mixup regularization [32], but regularizers like mixup are orthogonal to our approach and can be added to many
methods like ours (R5). Due to time constraints, we couldn't show its results in this rebuttal.

**Weight norms (R1, R5)**: Thanks to R1 for pointing to the recent development in the evolution of weight norms as a
result of gradient norms. We agree with R1 that weight norm analysis for the initial layers is complex due to batchnorm
and weight decay. However, we do see the effect of weight magnification on miscalibration, as also shown in Section C
and Fig C.1 of the Appendix, where we use a simple network without batchnorm or weight decay. We use weight decay
(L2 penalty) in other experiments (R1). The decrease in weight norm of the last layer up to epoch 150 can be attributed
to weight decay and batchnorm (R5), as also indicated by R1. Hence one may look at weight magnification *relative* to
other loss functions (see Fig. 2(e) in the main paper). As R5 expects, the feature norm of the last layer output (Figure
H.1 in Appendix) is increasing at least until epoch 150. We will clarify this in the paper and release the code.

**NLL+entropy regularization (R1)**: We did run NLL + entropy regularization [26] experiments, but found those
models to perform poorly for calibration, as the models were underconfident. However, please note that [26] shows that
label smoothing [19] is equivalent to this entropy regularization if the order of the KL divergence between uniform
distributions and model's outputs is reversed (Sec. 3.2 of [26]), and we show the results of label smoothing as a baseline.

**Threshold on $p_0$ (R5)**: We use Propositions 1,2 and Fig 3(a) to set $p_0$. We select the threshold $p_0$ such that $g(p_0, \gamma) = 1$
for a $\gamma$ that does not lead to dying gradients (high $\gamma$) or insufficient regularisation (low $\gamma$). $\gamma = 3$ proved to be a good
candidate and $g(0.25, 3) \approx 1$. Hence, $p_0 = 0.25$. We will clarify this.

**Table 1 (R2,R5)**: Thanks, we will move the vanilla focal loss results and accuracy results from the supplementary to
the main paper, using the extra page in the final version. We will also add the confidence intervals to Table 1.

**Citations (R1,R2,R3)**: We will cite 1508.05154 for AdaECE, although we use absolute differences instead of RMS
error. We will cite ensembling methods and other papers (R1,R3) for context and make the citations more readable.

**Imbalanced datasets (R1)**: Thanks for the suggestion. This would be an interesting experiment, but the calibration of
imbalanced datasets is a major research topic on its own. Out of necessity, we therefore leave it for future work.

**Figures, typo, clarity and real-life impact (R1, R2, R3, R5)**: Thanks, we will make the suggested changes to make
the figures more intuitive to read. We will replace Figure 4 with the CIFAR-100 results from the supplementary (Figure
K.1). We will also take into account all other suggestions when preparing the final version.

[Meta-Review · NeurIPS 2020]

This paper was reviewed by 4 reviewers and there was unanimous agreement that the paper should be accepted (scores ranging from "marginally above threshold" to "clear accept"). I agree with the reviewers and recommend the paper be accepted. All 4 reviewers provided quite detailed suggestions for improvement in the paper and I strongly recommend that the authors carefully take these suggestions into account in revising the paper for the camera-ready version. In particular, please be sure to please take into account the reviewer suggestions (R1 and R2 specifically) on improving and clarifying the wording of your OOD claims. And please also be sure to include vanilla Focal Loss results in the main paper (R1, R2, R5).